# TACKLING BYZANTINE CLIENTS IN FEDERATED LEARNING

## ABSTRACT

The possibility of adversarial (a.k.a., *Byzantine*) clients makes federated learning (FL) prone to arbitrary manipulation. The natural approach to robustify FL against adversarial clients is to replace the simple averaging operation at the server in the standard FedAvg algorithm by a *robust averaging rule*. While a significant amount of work has been devoted to studying the convergence of federated *robust averaging* (which we denote by FedRo), prior work has largely ignored the impact of *client subsampling* and *local steps*, two fundamental FL characteristics. While client subsampling increases the effective fraction of Byzantine clients, local steps increase the drift between the local updates computed by honest (i.e., non-Byzantine) clients. Consequently, a careless deployment of FedRo could yield poor performance. We validate this observation by presenting an in-depth analysis of FedRo with two-sided step-sizes, tightly analyzing the impact of client subsampling and local steps. Specifically, we present a sufficient condition on client subsampling for nearly-optimal convergence of FedRo (for smooth non-convex loss). Also, we show that the rate of improvement in learning accuracy *diminishes* with respect to the number of clients subsampled, as soon as the sample size exceeds a threshold value. Interestingly, we also observe that under a careful choice of step-sizes, the learning error due to Byzantine clients decreases with the number of local steps. We validate our theory by experiments on the FEMNIST image classification task.

## 1 INTRODUCTION

Federated Learning (FL) has emerged as a prominent machine-learning scheme since its inception by McMahan et al. (2017), thanks to its ability to train a model without centrally storing the training data (Kairouz et al., 2021). In FL, the training data is distributed across multiple machines, referred to as *clients*. The training procedure is coordinated by a central server that iteratively queries the clients on their local training data. The clients do not share their raw data with the server, hence reducing the risk of privacy infringement of training data usually present in traditional centralized schemes. The most common algorithm to train machine learning models in a federated manner is *federated averaging* (FedAvg) (McMahan et al., 2017). In short, at every round, the server holds a model that is broadcasted to a subset of clients, selected at random. Then, each selected client executes several steps of local updates using their local data and sends the resulting model back to the server. Upon receiving local models from the selected clients, the server averages them to update the global model.

However, training over decentralized data makes FL algorithms, such as FedAvg, quite vulnerable to adversarial, a.k.a., *Byzantine* (Lamport et al., 1982), clients that could sabotage the learning by sending arbitrarily bad local model updates. The problem of robustness to Byzantine clients in FL (and distributed learning at large) has received significant attention in the past (Blanchard et al., 2017; Data & Diggavi, 2021; Farhadkhani et al., 2022; Karimireddy et al., 2022; Gupta et al., 2023; Gorbunov et al., 2023; Zhu et al., 2023; Allouah et al., 2023; Guerraoui et al., 2023). The general idea for imparting robustness to an FL algorithm consists in replacing the averaging operation of the algorithm with a *robust aggregation rule*, basically seeking to filter out outliers. However, most previous works study the simplified FL setting where the server queries *all* clients, in every round, and each client only performs a *single* local update step. As of now, it remains unclear whether these results could readily apply to the standard FL setting (McMahan et al., 2017) where the server only queries a small subset of clients in each round, and each client performs multiple local updates. We

address this shortcoming of prior work by presenting an in-depth analysis of robust FL accounting for both *client subsampling* and *multiple local steps*. Our main findings are as follows.

**Key results.** We consider FedRo, a robust variant of FedAvg obtained by replacing the averaging operation at the server with *robust averaging*. We observe that it is impossible to ensure convergence of FedRo, in the presence of Byzantine clients, when the server samples a small number of clients, since the subset of clients sampled might contain a majority of Byzantine clients in some learning rounds with high probability. To circumvent this impossibility, we analyze a sufficient condition on the client subsampling size. Specifically, let $n$ and $b$ be the total number of clients and an upper bound on the number of Byzantine clients, respectively.[1] Consider $T$ learning rounds in which the server samples $\hat{n}$ clients. We show that the number of Byzantine clients sampled in each round is smaller than $\hat{b}$, with probability at least $p$, if

$$\hat{n} \geq \min \left\{ n, D\left(\frac{\hat{b}}{\hat{n}}, \frac{b}{n}\right)^{-1} \ln\left(\frac{T}{1-p}\right) \right\} , \qquad (1)$$

where $D(\alpha, \beta)$ denotes the Kullback–Leibler divergence between two Bernoulli distributions of respective parameters $\alpha$ and $\beta$. Under the above condition on the client subsampling size $\hat{n}$, we show that, ignoring higher order error terms in $T$, FedRo converges to an $\varepsilon$-stationary point with (refer to Corollary 1 for the complete expression)

$$\varepsilon \in \mathcal{O}\left( \sqrt{\frac{1}{\hat{n}T}\left[\frac{\sigma^2}{K} + \left(1 - \frac{\hat{n}-\hat{b}}{n-b}\right)\zeta^2\right]} + \frac{\hat{b}}{\hat{n}}\left(\frac{\sigma^2}{K} + \zeta^2\right) \right) , \qquad (2)$$

where $K$ is the number of local steps performed by each honest (i.e., non-Byzantine) client, $\sigma^2$ is the variance of the stochastic noise of the computed gradients and $\zeta^2$ measures the heterogeneity of the losses across the honest clients. The first term is similar to the one obtained for FedAvg with two-sided step-sizes and without the presence of Byzantine clients (Karimireddy et al., 2020), and it vanishes when $T \to \infty$. The second term is the additional non-vanishing error we incur due to Byzantine clients. Notably, our convergence analysis shows that the asymptotic error, in the presence of Byzantine clients, might improve upon increasing the number of local update steps executed by honest clients. This arises from the fact that an honest client's update is approximately the average of $K$ stochastic gradients, effectively reducing the variance of local updates while we maintain control over the deviation of local models through a careful choice of two-sided step-sizes. It is important to note that this observation does not contradict the findings of previous works (Karimireddy et al., 2020; Yang et al., 2021) regarding the additional error due to multiple local steps, as this error term exhibits a more favorable dependence on $T$ and is omitted in (2) (see Corollary 1).

We remark that condition (1) is quite intricate since it intertwines two tunable parameters of the algorithm, i.e., $\hat{n}$ and $\hat{b}$. We provide a practical approach to validate this condition. First, we show (by construction) that there exists a subsampling threshold $\hat{n}_{th}$ such that, for all $\hat{n} \geq \hat{n}_{th}$, there exists $\hat{b} < \hat{n}/2$ such that (1) is satisfied. Moreover, we show that increasing $\hat{n}$ further leads to a provable reduction in the asymptotic error. However, for another threshold value $\hat{n}_{opt}$, the rate of improvement in the asymptotic error with the sample size $\hat{n}$ *diminishes*, i.e., effectively saturates once $\hat{n}$ exceeds $\hat{n}_{opt}$. We refer to this phenomenon as *the diminishing return* of FedRo.

**Comparison with prior work.** As noted earlier, most prior work on Byzantine robustness in FL considers a simplified FL setting. Specifically, the impact of client subsampling is often overlooked. Moreover, clients are often assumed to take only a single local step, with the exception of the work by Gupta et al. (2023). The only existing work, to the best of our knowledge, that considers both client subsampling and multiple local steps is the work of Data & Diggavi (2021). However, we note that our convergence guarantee is significantly tighter on several fronts, expounded below.

In the absence of Byzantine clients, we recover the standard convergence rate of FedAvg (see Section 4.2). Whereas, the convergence rate of Data & Diggavi (2021) has a residual asymptotic error $\mathcal{O}(K\sigma^2 + K\zeta^2)$. Then, we show that the asymptotic error with Byzantine clients is inversely proportional to the number of local steps $K$ (see (2)), contrary to the aforementioned bound of Data &

---

[1] We assume that $b < n/2$, otherwise the goal of robustness is rendered vacuous (Liu et al., 2021).

Diggavi (2021) that is proportional to $K$. Moreover, while client subsampling is considered by Data & Diggavi (2021), it is assumed that Byzantine clients represent less than one-third of the subsampled clients in all learning rounds, which need not be true in general. Indeed, we prove that if the sample size $\hat{n}$ is small, then the fraction of subsampled Byzantine clients is larger than $1/2$ in some learning rounds with high probability (see Section 5.1). Lastly, our analysis applies to a broad spectrum of robustness schemes, unlike that of Data & Diggavi (2021), which only considers a specific scheme. In fact, the scheme considered in that paper has time complexity $\mathcal{O}(\min(d, \hat{n}) \cdot \hat{n}^2 d)$, where $d$ is the model size. Importantly, our findings eliminate the need for non-standard aggregation rules when incorporating local steps and client subsampling in Byzantine federated learning.

**Paper outline.** The rest of the paper is organized as follows. In Section 2, we formalize the problem of FL in the presence of Byzantine clients. Section 3 presents the FedRo algorithm and the property of robust averaging. In Section 4, we provide a theoretical analysis of the algorithm and give its convergence guarantees. In Section 5, we present a practical approach for choosing the subsampling size and show the diminishing return of the algorithm. Lastly, in Section 6, we validate our theoretical results through numerical experiments.

## 2 PROBLEM STATEMENT

We consider the standard FL setting comprising a server and $n$ clients, represented by set $[n] := \{1, \ldots, n\}$ and a scenario where at most $b$ (out of the total $n$) clients may be adversarial[2], i.e., Byzantine. We define $\mathcal{B} \subset [n]$ as the associated subset containing the Byzantine clients and $\mathcal{H} \subset [n]$ the subset of honest clients such that $\mathcal{H} \cup \mathcal{B} = [n]$ and $\mathcal{H} \cap \mathcal{B} = \emptyset$. Consider a data space $\mathcal{Z}$ and a differentiable loss function $f : \mathbb{R}^d \times \mathcal{Z} \to \mathbb{R}$. Given a parameter $x \in \mathbb{R}^d$, a data point $\xi \in \mathcal{Z}$ incurs a loss of $f(x, \xi)$. For each honest client $i \in \mathcal{H}$, we consider a local data distribution $\mathcal{D}^{(i)}$ and its associated local loss function $f^{(i)}(x) := \mathbb{E}_{\xi \sim \mathcal{D}^{(i)}}[f(x, \xi)]$. The goal is to minimize the global loss function defined by the average loss function over all honest clients (Karimireddy et al., 2022):

$$F(x) := \frac{1}{|\mathcal{H}|} \sum_{i \in \mathcal{H}} f^{(i)}(x) \ , \tag{3}$$

More precisely, the goal, formally defined below, is to find an $\varepsilon$-stationary point of the global loss function in the presence of Byzantine clients.

**Definition 1 ($(n, b, \varepsilon)$-Byzantine resilience).** *A federated learning algorithm is $(n, b, \varepsilon)$-Byzantine resilient if, despite the presence of at most $b$ Byzantine clients out of $n$ clients, it outputs $\hat{x}$ such that*

$$\mathbb{E}\left[\|\nabla F(\hat{x})\|^2\right] \leq \varepsilon \ ,$$

*where the expectation is over the randomness of the algorithm.*

**Assumptions.** We make the following three assumptions. The first two assumptions are standard in first-order stochastic optimization (Ghadimi & Lan, 2013; Bottou et al., 2018). The third assumption is essential for obtaining a meaningful Byzantine robustness guarantee (Allouah et al., 2023).

**Assumption 1 (Smoothness).** *There exists $L < \infty$ such that $\forall i \in \mathcal{H}$, and $\forall x, y \in \mathbb{R}^d$, we have*

$$\left\|\nabla f^{(i)}(x) - \nabla f^{(i)}(y)\right\| \leq L \|x - y\| \ .$$

Note that for all $i \in \mathcal{H}$, as $f$ is a differentiable function, by definition of $f^{(i)}(x)$, we have

$$\mathbb{E}_{\xi \sim \mathcal{D}^{(i)}}[\nabla f(x, \xi)] = \nabla f^{(i)}(x) \ .$$

**Assumption 2 (Bounded local noise).** *There exists $\sigma < \infty$ such that $\forall i \in \mathcal{H}$, and $\forall x \in \mathbb{R}^d$, we have*

$$\mathbb{E}_{\xi \sim \mathcal{D}^{(i)}}\left[\left\|\nabla f(x, \xi) - \nabla f^{(i)}(x)\right\|^2\right] \leq \sigma^2 \ .$$

**Assumption 3 (Bounded heterogeneity).** *There exists $\zeta < \infty$ such that for all $x \in \mathbb{R}^d$, we have*

$$\frac{1}{|\mathcal{H}|} \sum_{i \in \mathcal{H}} \left\|\nabla f^{(i)}(x) - \nabla F(x)\right\|^2 \leq \zeta^2.$$

---

[2]Considering a known upper bound on the total number of Byzantine clients is standard and essential for obtaining tight convergence guarantees (Blanchard et al., 2017; Karimireddy et al., 2022; Allouah et al., 2023).

## 3  ROBUST FEDERATED LEARNING

We present here the natural Byzantine-robust adaptation of FedAvg. In FedAvg, the server iteratively updates the global model using the *average* of the partial updates sent by the clients. However, averaging can be manipulated by a single Byzantine client (Blanchard et al., 2017). Thus, in order to impart robustness to the algorithm, previous work often replaces averaging with a robust aggregation rule $\mathcal{A}$. We call this robust variant *Federated Robust averaging* (or FedRo).

**Robust aggregation rule.** At the core of FedRo is the robust aggregation function $\mathcal{A}$ that provides a good estimate of the honest clients' updates in the presence of Byzantine updates. Notable robust aggregation functions are Krum (Blanchard et al., 2017), geometric median (Pillutla et al., 2022; Acharya et al., 2022), mean-around-median (MeaMed) (Xie et al., 2018), coordinate-wise median and trimmed mean (Yin et al., 2018b), and minimum diameter averaging (MDA) (El-Mhamdi et al., 2021). Considering a total number of $\hat{n}$ input vectors, most aggregation rules are designed under the assumption that the maximum number of Byzantine vectors is upper bounded by a given parameter $\hat{b} < \hat{n}/2$. To formalize the robustness of an aggregation rule, we use the robustness criterion of $(\hat{n}, \hat{b}, \kappa)$-robustness, introduced by Allouah et al. (2023). This criterion is satisfied by most existing aggregation rules, and allows obtaining optimal convergence guarantees. Formally, $(\hat{n}, \hat{b}, \kappa)$-robustness is defined as follows.

**Definition 2** ($(\hat{n}, \hat{b}, \kappa)$**-robustness (Allouah et al., 2023)**)**.** *Let $\hat{b} < \hat{n}/2$ and $\kappa \geq 0$. An aggregation rule $\mathcal{A} \colon \mathbb{R}^{d \times \hat{n}} \to \mathbb{R}^d$ is said to be $(\hat{n}, \hat{b}, \kappa)$-robust, if for any vectors $w_1, \dots, w_{\hat{n}} \in \mathbb{R}^d$, and any subset $S \subseteq [\hat{n}]$ of size $\hat{n} - \hat{b}$, we have*

$$\left\| \mathcal{A}\left(w_1, \dots, w_{\hat{n}}\right) - \overline{w}_S \right\|^2 \leq \frac{\kappa}{|S|} \sum_{i \in S} \left\| w_i - \overline{w}_S \right\|^2, \quad where \quad \overline{w}_S = \frac{1}{|S|} \sum_{i \in S} w_i \ .$$

---

**Algorithm 1:** FedRo: FedAvg with a robust aggregation rule $\mathcal{A}$

**Input**  : Initial model $x_0$, number of rounds $T$, number of local steps $K$, client step-size $\gamma_c$, server step-size $\gamma_s$, sample size $\hat{n}$, and tolerable number of Byzantine clients $\hat{b}$.

**for** $t = 0$ *to* $T - 1$ **do**
 **Server** selects a subset $\mathcal{S}_t$ of $\hat{n}$ clients, uniformly at random from the set of all clients $[n]$, and broadcasts $x_t$ to them;
 **for** *each* **honest client** $i \in \mathcal{S}_t$ *(in parallel)* **do**
  Set $x_{t,0}^{(i)} = x_t$;
  **for** $k \in \{0, \dots, K-1\}$ **do**
   Sample a data point $\xi_{t,k}^{(i)} \sim \mathcal{D}^{(i)}$, compute a stochastic gradient $g_{t,k}^{(i)} := \nabla f(x_{t,k}^{(i)}, \xi_{t,k}^{(i)})$
   and perform $x_{t,k+1}^{(i)} := x_{t,k}^{(i)} - \gamma_c g_{t,k}^{(i)}$;
  Send to the server $u_t^{(i)} := x_{t,K}^{(i)} - x_t$;

 (A Byzantine client $i$ may send an arbitrary value for $u_t^{(i)}$ to the server.)
 **Server** updates the parameter vector $x_{t+1} = x_t + \gamma_s \mathcal{A}\left(u_t^{(i)}, i \in \mathcal{S}_t\right)$;

**Output :** $\hat{x}$ selected uniformly at random from $\left\{x_0, \dots, x_{T-1}\right\}$.

---

**Description of** FedRo **as presented in Algorithm 1.** The server starts by initializing a model $x_0$. Then at each round $t \in \{0, \dots, T-1\}$, it maintains a parameter vector $x_t$ which is broadcast to a subset $\mathcal{S}_t$ of $\hat{n}$ clients selected uniformly at random from $[n]$ without replacement. Among those $\hat{n}$ selected clients, some of them might be Byzantine. We assume that at most $\hat{b}$ out of the $\hat{n}$ clients are Byzantine where $\hat{b}$ is a parameter given as an input to the algorithm. We show later in sections 4 and 5 how to set $\hat{n}$ and $\hat{b}$ such that this assumption holds with high probability. Each honest client $i \in \mathcal{S}_t$ updates its current local model as $x_t^{(i,0)} = x_t$. The round proceeds in three phases:

- *Local computations.* Each honest client $i \in \mathcal{S}_t$ performs $K$ successive local updates where each local update $k \in [K]$ consists in sampling a data point $\xi_{t,k}^{(i)} \sim \mathcal{D}^{(i)}$, computing a stochastic

gradient $g_{t,k}^{(i)} := \nabla f(x_{t,k}^{(i)}, \xi_{t,k}^{(i)})$ and making a local step:

$$x_{t,k+1}^{(i)} := x_{t,k}^{(i)} - \gamma_c g_{t,k}^{(i)} \ ,$$

where $\gamma_c$ is the client step-size.

- *Communication phase.* Each honest client $i \in \mathcal{S}_t$ computes the difference between the global model and its own model after $K$ local updates as:

$$u_t^{(i)} := x_{t,K}^{(i)} - x_t,$$

and sends it to the server.

- *Global model update.* Upon receiving the update vectors from all the selected clients (Byzantine included) in $\mathcal{S}_t$, the server updates the global model using an $(\hat{n}, \hat{b}, \kappa)$-robust aggregation rule $\mathcal{A} : \mathbb{R}^{\hat{n} \times d} \to \mathbb{R}^d$. Specifically, the global update can be written as follows

$$x_{t+1} = x_t + \gamma_s \mathcal{A}\left(u_t^{(i)}, i \in \mathcal{S}_t\right) \ ,$$

where $\gamma_s$ is called the server step-size.

## 4 THEORETICAL ANALYSIS

In this section, we present a detailed theoretical analysis of Algorithm 1. We first present a sufficient condition on the parameters $\hat{b}$ and $\hat{n}$ for the convergence of FedRo. Then, we present our main result demonstrating the convergence of FedRo in the presence of at most $b$ Byzantine clients. Finally, we explain how this result matches our original goal of designing a $(n, b, \varepsilon)$-Byzantine resilient algorithm, and highlight the impact of local steps on the robustness of the scheme.

### 4.1 SUFFICIENT CONDITION ON $\hat{n}$ AND $\hat{b}$

We first make a simple observation by considering the sampling of only one client at each round, i.e., $\hat{n} = 1$. Then, convergence can only be ensured if this client is non-Byzantine in all $T$ rounds of training. Indeed, if a Byzantine client is sampled, it gets full control of the learning process, compromising any possibility for convergence. Accordingly, we can have convergence only with probability $(1 - b/n)^T$, which decays exponentially in $T$. This simple observation indicates that having an excessively small sample size can be dangerous in the presence of Byzantine clients. More generally, recall that the aggregation function requires a parameter $\hat{b}$, which is an upper bound on the number of Byzantine clients in the sample. If, at a given round, we have more than $\hat{b}$ in the sample, Byzantine clients can control the output of the aggregation and we cannot provide any learning guarantee. Specifically, denoting by $\hat{\mathcal{B}}_t$ the set of Byzantine clients selected at round $t$, the learning process can only converge if the following event holds true:

$$\mathcal{E} = \left\{ \forall t \in \{0, \ldots, T-1\}, \left|\hat{\mathcal{B}}_t\right| \leq \hat{b} \right\} \ . \tag{4}$$

That is, in all rounds, the number of selected Byzantine clients is at most $\hat{b}$. This motivates us to devise a condition on $\hat{n}$ and $\hat{b}$ in FedRo that will be sufficient for $\mathcal{E}$ to hold with high probability. This condition is provided in Lemma 1 below, considering the non-trivial case where $b > 0$.

**Lemma 1.** *Let $p < 1$ and $b$ be such that $0 < b/n < 1/2$. Consider* FedRo *as defined in Algorithm 1. Suppose that $\hat{n}$ and $\hat{b}$ are such that $b/n < \hat{b}/\hat{n} < 1/2$ and*

$$\hat{n} \geq \min\left\{ n, D\left(\frac{\hat{b}}{\hat{n}}, \frac{b}{n}\right)^{-1} \ln\left(\frac{T}{1-p}\right) \right\} \ , \tag{5}$$

*with $D(\alpha, \beta) := \alpha \ln(\alpha/\beta) + (1-\alpha) \ln(1-\alpha/1-\beta)$, for $\alpha, \beta \in (0, 1)$. Then, Event $\mathcal{E}$ as defined in (4) holds true with probability at least $p$.*

Lemma 1 presents a sufficient condition on $\hat{n}$ and $\hat{b}$ to ensure that, with probability at least $p$, the server does not sample more Byzantine clients than the expected number $\hat{b}$. Note, however, that this condition is non-trivial to satisfy due to the convoluted dependence between $\hat{b}$ and $\hat{n}$. We defer to Section 5 an explicit strategy to choose both $\hat{n}$ and $\hat{b}$ so that (5) holds. In the next section, we will first demonstrate the convergence of FedRo given this condition.

### 4.2 Convergence of FedRo

We now present the convergence analysis of FedRo in Theorem 1 below. Essentially, we consider Algorithm 1 with sufficiently small constant step-sizes $\gamma_c$ and $\gamma_s$, and when assumptions 1, 2, and 3 hold true. We show that, with high probability, FedRo achieves a training error similar to FedAvg (Karimireddy et al., 2020), plus an additional error term due to Byzantine clients.

---

**Theorem 1.** *Consider Algorithm 1. Suppose assumptions 1, 2, and 3 hold true. Let $p < 1$, and assume $\hat{n}$ and $\hat{b}$ are such that $b/n < \hat{b}/\hat{n} < 1/2$ and (5) hold. Suppose $\mathcal{A} \colon \mathbb{R}^{d \times \hat{n}} \to \mathbb{R}^d$ is a $(\hat{n}, \hat{b}, \kappa)$-robust aggregation rule and the step-sizes are such that $\gamma_c \leq 1/16LK$ and $\gamma_c\gamma_s \leq 1/36LK$. Then, with probability at least $p$ we have,*

$$
\frac{1}{T} \sum_{t=0}^{T-1} \mathbb{E}\left[\|\nabla F(x_t)\|^2\right] \leq \frac{5}{TK\gamma_c\gamma_s}\Delta_0 + \frac{20LK\gamma_c\gamma_s}{\hat{n}}\left(\frac{\sigma^2}{K} + 6\left(1 - \frac{\hat{n}-\hat{b}}{n-b}\right)\zeta^2\right)
$$
$$
+ \frac{10}{3}\gamma_c^2 L^2(K-1)\sigma^2 + \frac{40}{3}\gamma_c^2 L^2 K(K-1)\zeta^2 + 165\kappa\left(\frac{\sigma^2}{K} + 6\zeta^2\right),
$$

*where $\Delta_0$ is a real value such that $F(x_0) - \min_{x \in \mathbb{R}^d} F(x) \leq \Delta_0$.*

---

Setting $\gamma_c = 1$, $\hat{n} = n$, $\hat{b} = b$ and $K = 1$, FedRo reduces to the robust implementation of distributed SGD studied by several previous works (Blanchard et al., 2017; Yin et al., 2018b). In this case, the last term of our convergence guarantee is in $O\left(\kappa(\sigma^2 + \zeta^2)\right)$, which is an additive factor $\mathcal{O}(\kappa\sigma^2)$ away from the optimal bound (Allouah et al., 2023; Karimireddy et al., 2022). However, the dependency on $\sigma$ for FedRo is unavoidable because the algorithm does not use noise reduction techniques, as prescribed by Karimireddy et al. (2021); Farhadkhani et al. (2022); Gorbunov et al. (2023). These techniques typically rely on the observation that, when a client computes stochastic gradients over several rounds, one can construct update vectors with decreasing variance by utilizing previously computed stochastic gradients. However, in a practical federated learning setting, when we typically have $n \gg \hat{n}$, individual clients may only be selected for a limited number of rounds, rendering it impractical to rely on previously computed gradients to reduce the variance. It thus remains open to close the $\mathcal{O}(\kappa\sigma^2)$ gap between the upper and lower bounds in the federated setting.

**Byzantine resilience of** FedRo. Using Theorem 1, we can show that Algorithm 1 guarantees $(n, b, \varepsilon)$-resilience. In doing so, we first need to select step-sizes that would both simplify the upper bound and respect the conditions of Theorem 1. One suitable choice of step-sizes is $\gamma_s = 1$ and

$$
\gamma_c = \min\left\{\frac{1}{36LK}, \frac{1}{2K}\sqrt{\frac{\hat{n}\Delta_0}{LT\left(\frac{\sigma^2}{K} + 6\left(1 - \frac{\hat{n}-\hat{b}}{n-b}\right)\zeta^2\right)}}, \sqrt[3]{\frac{3\Delta_0}{2K(K-1)TL^2(\sigma^2 + 4K\zeta^2)}}\right\}. \quad (6)
$$

Using the above step-sizes, we obtain the following corollary on the resilience of FedRo.

**Corollary 1.** *Under the same conditions as Theorem 1, denoting $r := 1 - \frac{\hat{n}-\hat{b}}{n-b}$, Algorithm 1 with $\gamma_s = 1$ and $\gamma_c$ as defined in (6) guarantees $(n, b, \varepsilon)$-Byzantine resilience, with high probability, for*

$$
\varepsilon \in \mathcal{O}\left(\sqrt{\frac{L\Delta_0}{\hat{n}T}\left(\frac{\sigma^2}{K} + r\zeta^2\right)} + \sqrt[3]{\frac{(1-\frac{1}{K})L^2\Delta_0^2(\frac{\sigma^2}{K}+\zeta^2)}{T^2}} + \frac{L\Delta_0}{T} + \kappa\left(\frac{\sigma^2}{K} + \zeta^2\right)\right).
$$

The first three terms in Corollary 1 are similar to the convergence rate we would obtain for FedAvg without Byzantine clients (Karimireddy et al., 2020). These terms vanish when $T \to \infty$ with a leading term in $\mathcal{O}\left(1/\sqrt{\hat{n}T}\right)$. Hence FedRo preserves the linear speedup in $\hat{n}$ of FedAvg (Yang et al., 2021). Moreover, similar to the non-Byzantine case (Karimireddy et al., 2020), employing multiple local steps (i.e., $K > 1$) introduces an additional bias, resulting in the second error term in Corollary 1 which is in $\mathcal{O}(T^{-2/3})$. On the other hand, the last term in Corollary 1 is the additional error caused by Byzantine clients, and does not decrease with $T$. Using an aggregation function with[3] $\kappa \in \mathcal{O}(\hat{b}/\hat{n})$,

---

[3]This is the optimal value for $\kappa$ as shown by Allouah et al. (2023).

such as coordinate-wise trimmed mean (Allouah et al., 2023), this term will be in

$$\mathcal{O}\left(\frac{\hat{b}}{\hat{n}}\left(\frac{\sigma^2}{K}+\zeta^2\right)\right) \quad . \tag{7}$$

Intuitively, the non-vanishing term $\sigma^2/K$ quantifies the uncertainty due to the stochastic noise in gradient computations, which may be exploited by Byzantine clients at each aggregation step. Interestingly, this uncertainty decreases by increasing the number of local steps $K$, for a sufficiently small local step-size. This is because, intuitively, with more local steps, we are effectively approximating the average of a larger number of stochastic gradients. This shows the advantage of having multiple local steps for federated learning in the presence of Byzantine clients. The same benefit of local steps has been observed in other constrained learning problems, e.g., in the trade-off between utility and privacy in distributed differentially private learning (Bietti et al., 2022). Note that the error term in (7) also features a multiplicative term $\hat{b}/\hat{n}$. Hence, we can reduce the asymptotic error by carefully selecting the parameters $\hat{n}$ and $\hat{b}$ satisfying condition (5) such that $\hat{b}/\hat{n}$ is minimized. We present in the next section an order-optimal methodology for choosing these parameters.

## 5 On the Choice of $\hat{b}$ and $\hat{n}$

Condition (5) involves a complex interplay between parameters $\hat{n}$ and $\hat{b}$. Recall that the value of $\hat{n}$ affects the communication complexity of FedRo, while the fraction $\hat{b}/\hat{n}$ controls the upper bound on the asymptotic error in (7). Thus, it is primary to understand how to set $\hat{n}$ and $\hat{b}$ while minimizing the complexity overhead and optimizing the error. First, note that for a fixed $\hat{n}$, the optimal choice of $\hat{b}$ is given as a solution to the following optimization problem.

$$\min_{\hat{b}\in\left(\frac{b}{n}\hat{n},\frac{1}{2}\hat{n}\right)}\left\{\hat{b}\quad\text{subject to}\quad D\left(\frac{\hat{b}}{\hat{n}},\frac{b}{n}\right)\geq\frac{1}{\hat{n}}\ln\left(\frac{T}{1-p}\right)\right\} \quad . \tag{8}$$

We denote the minimizer by $\hat{b}_\star$. In essence, we aim to select the smallest $\hat{b}$ within the interval $\left(\frac{b}{n}\hat{n},\frac{1}{2}\hat{n}\right)$ that satisfies condition (5), thereby minimizing the asymptotic error in (7). Notably, since $\hat{b}$ is an integer ranging from $\lceil b/n\cdot\hat{n}\rceil$ to $\lfloor\hat{n}/2\rfloor$, and the second inequality is monotonic within this range, we can efficiently compute the solution to (8) in $\mathcal{O}(\log\hat{n})$ steps using binary search.

The selection of $\hat{n}$ is more involved, as we aim to satisfy a number of potentially conflicting objectives.

- On one hand, $\hat{n}$ must be sufficiently large to ensure the convergence of FedRo. In Section 5.1, we introduce a threshold value $\hat{n}_{th}$ such that, for any $\hat{n}\geq\hat{n}_{th}$, the feasible region of (8) remains non-empty, thereby guaranteeing the convergence of FedRo by Theorem 1. We also show that the value of $\hat{n}_{th}$ is nearly optimal in this context.

- On the other hand, while increasing $\hat{n}$ improves the asymptotic error in (7), it also induces more communication overhead to the system. We show, in Section 5.2, the existence of a second threshold, denoted as $\hat{n}_{opt}$, which lies above $\hat{n}_{th}$. Beyond this threshold, FedRo attains an order-optimal error with respect to the fraction of Byzantine clients. Thus, increasing $\hat{n}$ beyond $\hat{n}_{opt}$ does not lead to any further improvement in the asymptotic error of FedRo.

### 5.1 Sample size threshold for convergence

We begin by presenting Lemma 2, which establishes a lower bound $\hat{n}_{th}$ on the sample size necessary to ensure the convergence of FedRo:

**Lemma 2.** *Consider the sampling threshold $\hat{n}_{th}$ defined as follows*

$$\hat{n}_{th}:=\left\lceil D\left(\frac{1}{2},\frac{b}{n}\right)^{-1}\ln\left(\frac{4T}{1-p}\right)\right\rceil+2 \quad , \tag{9}$$

*with D as defined in Lemma 1. If $\hat{n}\geq\hat{n}_{th}$, then there exist $\hat{b}$ such that (5) holds and $b/n<\hat{b}/\hat{n}<1/2$.*

Note that $\hat{n}_{th}$ as defined in Lemma 2 is tight, in the sense that for any $\hat{n}$ smaller than $\hat{n}_{th}$ by more than a constant multiple, any algorithm which relies on sampling clients uniformly at random will fail to converge with non-negligible probability, i.e., $\neg\mathcal{E}$ holds with probability greater than $1 - p$.

**Lemma 3.** *Let $p \geq 1/2$ and suppose that the fraction $b/n < 1/2$ is a constant. Consider* FedRo *as defined in Algorithm 1 with*

$$\hat{n} < \left( D\left( \frac{1}{2}, \frac{b}{n} \right) + 2 \right)^{-1} \ln\left( \frac{T}{3(1-p)} \right) - 1 \; ,$$

*where $D$ is defined in Lemma 1. Then for large enough $n$ and any $\hat{b} < \hat{n}/2$, Event $\mathcal{E}$ as defined in (4) holds true with probability strictly smaller than $p$.*

## 5.2 SAMPLE SIZE THRESHOLD FOR ORDER-OPTIMAL ERROR

Now note that increasing $\hat{n}$ further beyond $\hat{n}_{th}$ enables us to decrease the fraction $\hat{b}/\hat{n}$ leading to a better convergence guarantee in (7). Previous works (Karimireddy et al., 2021; 2022) suggest that the asymptotic error is lower bounded by[4] $\Omega(b/n(\sigma^2/K + \zeta^2))$. As such, we aim to find the values of $\hat{n}$ for which we can obtain an asymptotic error in $\mathcal{O}(b/n(\sigma^2/K + \zeta^2))$. In Lemma 4 below, we provide a sufficient condition on the sampling parameter for such a convergence rate to hold.

**Lemma 4.** *Suppose we have $0 < \frac{b}{n} < \frac{1}{2}$ and consider the sampling threshold $\hat{n}_{opt}$ defined as follows*

$$\hat{n}_{opt} := \left\lceil \max\left\{ \frac{1}{(1/2 - b/n)^2}, \frac{3}{b/n} \right\} \ln\left( \frac{4T}{1-p} \right) \right\rceil + 2 \; . \tag{10}$$

*If $\hat{n} \geq \hat{n}_{opt}$, then, the solution $\hat{b}_\star$ to (8) exists and satisfies $\hat{b}_\star/\hat{n} \in \mathcal{O}\left( b/n \right)$.*

**Remark 1.** *Note that in Lemma 2 (resp. Lemma 4), we may set $\hat{n}_{th}$ (resp. $\hat{n}_{opt}$) to $n$ if the quantity on the right hand side exceeds $n$ without violating the conclusion of the lemma.*

**Diminishing returns.** Combining (7) and Lemma 4, we conclude that for $\hat{n} \geq \hat{n}_{opt}$, FedRo achieves the order-optimal asymptotic error. Notably, this implies that the performance improvement *saturates* when $\hat{n} = \hat{n}_{opt}$. Beyond this point, further increases in $\hat{n}$ do not affect the error order. In other words, we can obtain the best asymptotic error (leading term in convergence guarantee) in Byzantine federated learning without sampling all clients. This contrasts with (non-Byzantine) federated learning where often the leading term is in $\mathcal{O}(1/\sqrt{\hat{n}T})$, and always improves by increasing $\hat{n}$.

## 6 EMPIRICAL RESULTS

In this section, we first illustrate the dependencies of $\hat{n}_{th}$ and $\hat{n}_{opt}$ to the global fraction of Byzantine clients $b/n$ and show the diminishing return of FedRo. In a second step, we demonstrate the existence of an empirical threshold on the number of subsampled clients, below which FedRo does not converge. Next, we show the performance of FedRo with chosen values of $\hat{n}$ and $\hat{b}$ as described in Section 5. Finally, we evaluate the impact of the number of local steps on the performance of FedRo. For more details on the different configurations, refer to Appendix C.

**Understanding the diminishing return.** First, we study the variation of $\hat{n}_{th}$ and $\hat{n}_{opt}$ with respect to the fraction of Byzantine clients $b/n$ using (9) and (10) respectively. By setting the number of rounds $T$ to 500, the probability of success $p$ to 0.99 and the number of clients $n$ to 1000, we observe in Figure 1 that in some regimes, for example when the fraction of Byzantine clients is about 0.2, sampling only 20% of the clients allows converging and gives an order-optimal error.

**Existence of an empirical threshold.** In a second experiment, we train a convolutional neural network (CNN) on a portion of the FEMNIST dataset (Caldas et al., 2018) using FedRo over $T = 500$ rounds. We consider $n = 150$ clients where 30 of them are Byzantine, hence $b = 30$. We use different values of $\hat{n}$ to emphasize the need for subsampling beyond a threshold and choose

---

[4]More precisely, Karimireddy et al. (2021) prove the lower bound with respect to $\sigma$ for a family of the algorithms called permutation invariant (which includes FedAvg and Algorithm 1). Even though the lower bound is proved without considering local steps, it can be easily generalized to this case.

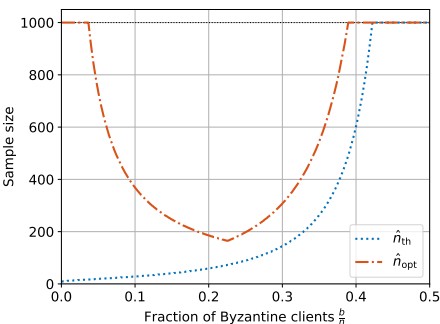

Figure 1: Variation of $\hat{n}_{th}$ and $\hat{n}_{opt}$ with respect to the fraction of Byzantine clients.

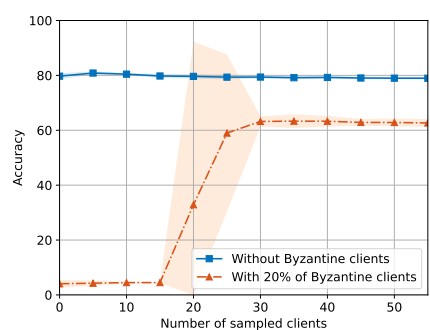

Figure 2: Accuracy of FedRo with respect to the number of subsampled clients.

$\hat{b} = \lfloor \hat{n}/2 \rfloor$. Here we use the state-of-the-art NNM robustness scheme (Allouah et al., 2023) coupled with coordinate-wise trimmed-mean (Yin et al., 2018a) and observe that FedRo fails when the number of sampled clients is below 30. This validates our theoretical result in Lemma 3.

**Empirical performance of** FedRo**.** Next, we use our results from Section 5 to run FedRo on a CNN with appropriate and optimal values of $\hat{n}$ and $\hat{b}$. Specifically, we run the algorithm for 500 rounds on a portion of the FEMNIST dataset (Caldas et al., 2018) with $n = 150$ and $b = 15$. We compute the minimum number of clients $\hat{n}_{th}$ that we need to sample per round to have a guarantee of convergence with probability $p = 0.99$ and obtain $\hat{n}_{th} = 26$. For $\hat{n} = 26$, using (8), we compute the optimal value of $\hat{b}_\star = 11$ and thus fix $\hat{b} = 11$. To evaluate FedRo, we use different Byzantine attacks, namely *sign flipping* (SF) (Allen-Zhu et al., 2020), *fall of empires* (FOE) (Xie et al., 2019), *a little is enough* (ALIE) (Baruch et al., 2019) and *mimic* (Karimireddy et al., 2022). We present the results in Figure 3.

**Impact of the number of local steps.** In the same configuration as the prior experiment, we vary $K$ from 5 to 35 and assess FedRo against the same set of attacks. The local step size $\gamma_c$ is set as $\Theta(1/K)$ as suggested by Corollary 1. There are $n = 150$ total clients, with 30 being Byzantine ($b = 30$). Figure 4 illustrates that increasing the number of local steps $K$ enhances model accuracy against stronger attacks, validating our theoretical worst-case guarantees.

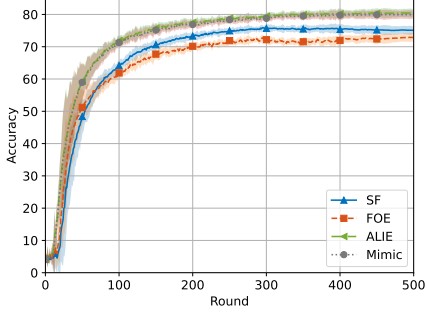

Figure 3: Accuracy of FedRo with NNM and Trimmed Mean on the FEMNIST dataset.

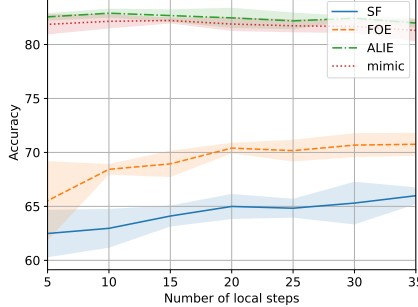

Figure 4: Accuracy of FedRo with respect to the number of local steps on FEMNIST.

# 7 CONCLUSION

We address the challenge of robustness to Byzantine clients in federated learning by analyzing FedRo, a robust variant of FedAvg. We precisely characterize the impact of client subsampling and multiple local steps, standard features of FL, often overlooked in prior work, on the convergence of FedRo. Our analysis shows that when the sample size is sufficiently large, FedRo achieves near-optimal convergence error that reduces with the number of local steps. For client subsampling, we demonstrate (i) the importance of tuning the client subsampling size appropriately to tackle the Byzantine clients, and (ii) the phenomenon of *diminishing return* where increasing the subsampling size beyond a certain threshold yields only marginal improvement on the learning error.

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

# Appendix

ORGANIZATION

The appendices are organized as follows:

- In Appendix A, we analyze the convergence of FedRo.
- In Appendix B, we discuss the results regarding the subsampling process in FedRo.
- In Appendix C, we provide the details about the experimental results of the paper.

## A  CONVERGENCE PROOF

In this section, we prove Theorem 1 showing the convergence of Algorithm 1.

For any set $A$, and integer $a$, we denote by $\mathcal{U}^a(A)$, the uniform distribution over all subsets of $A$ of size $a$. Recall that at each round $t \in \{0, \ldots, T-1\}$, the server selects a set $\mathcal{S}_t \sim \mathcal{U}^{\hat{n}}([n])$ of $\hat{n}$ clients uniformly at random from $[n]$ without replacement. Recall also that under event $\mathcal{E}$, defined in (4), the number of selected Byzantine clients at all rounds is bounded by $\hat{b}$. As we show that, under the condition stated in Theorem 1, this event holds with high probability (Lemma 1), to simplify the notation, throughout this section, we implicitly assume that event $\mathcal{E}$ holds and we assume it is given in all subsequent expectations. Now denoting by $\mathcal{H}_t^*$, the set of selected honest clients at round $t$, given event $\mathcal{E}$, the cardinality of $\mathcal{H}_t^*$ is at least $\hat{h} := \hat{n} - \hat{b}$. We then define $\hat{\mathcal{H}}_t \sim \mathcal{U}^{\hat{h}}(\mathcal{H}_t^*)$, a randomly selected subset of $\mathcal{H}_t^*$ of size $\hat{h}$. As (by symmetry) any two sets $\mathcal{H}^{(1)} \subset \mathcal{H}$, and $\mathcal{H}^{(2)} \subset \mathcal{H}$ of size $\hat{h}$ have the same probability of being selected, we have $\hat{\mathcal{H}}_t \sim \mathcal{U}^{\hat{h}}(\mathcal{H})$. Our convergence proof relies on analyzing the trajectory of the algorithm considering the average of the updates from the honest clients $\hat{\mathcal{H}}_t$, while taking into account the additional error that is introduced by the Byzantine clients.

### A.1  BASIC DEFINITIONS AND NOTATIONS

Let us denote by

$$G_t^{(i)} := \frac{1}{K} \sum_{k=0}^{K-1} g_{t,k}^{(i)} = -\frac{u_t^{(i)}}{K\gamma_c}, \tag{11}$$

the average of stochastic gradients computed by client $i$ along their trajectory at round $t$, where in the second equality we used the facts that $u_t^{(i)} := x_{t,K}^{(i)} - x_{t,0}^{(i)}$ from Algorithm 1. Moreover, we denote by

$$\Delta_t^{(i)} := \frac{1}{K} \sum_{k=0}^{K-1} \nabla f^{(i)}(x_{t,k}^{(i)}), \tag{12}$$

the average of the true gradients along the trajectory of client $i$ at round $t$. Also, denote by

$$e_t := \mathcal{A}\left(u_t^{(i)}, i \in \mathcal{S}_t\right) - \frac{1}{\hat{h}} \sum_{i \in \hat{\mathcal{H}}_t} u_t^{(i)}, \tag{13}$$

the difference between the output of the aggregation rule and the average of the honest updates.

We denote by $\mathcal{P}_t$ all the history up to the computation of $x_t$. Specifically,

$$\mathcal{P}_t := \{x_0, \ldots, x_t\} \cup \left\{ x_{s,k}^{(i)} : \forall k \in [K], \forall i \in [n], \forall s \in \{0 \ldots, t-1\} \right\} \cup \{\mathcal{S}_0 \ldots \mathcal{S}_{t-1}\}.$$

We denote by $\mathbb{E}_t[\cdot]$ and $\mathbb{E}[\cdot]$ the conditional expectation $\mathbb{E}[\cdot \mid \mathcal{P}_t]$ and the total expectation, respectively. Thus, $\mathbb{E}[\cdot] = \mathbb{E}_0[\cdots \mathbb{E}_T[\cdot]]$. Note also for computing $x_{t+1}$ from $x_t$, we have two sources of randomness: one for the set of clients that are selected and one for the random trajectory of each honest client. To obtain a tight bound, we sometimes require to take the expectation with respect to each of these two randomnesses separately. In this case, by the tower rule, we have

$\mathbb{E}_t [\cdot] = \mathbb{E}_{\hat{\mathcal{H}}_t \sim \mathcal{U}^{\hat{h}}(\mathcal{H})} \left[ \mathbb{E}_t \left[ \cdot | \hat{\mathcal{H}}_t \right] \right]$. Finally, for any $k \in \{0, \ldots, K-1\}$, and $t \in \{0, \ldots, T-1\}$ we denote by

$$\mathcal{P}_{t,k} := \mathcal{P}_t \cup \left\{ \forall i \in [n] : x_{t,0}^{(i)}, \ldots, x_{t,k}^{(i)}; \mathcal{S}_t \right\},$$

the history of the algorithm up to the computation of $x_{t,k}^{(i)}$, the $k$-th local update of the clients at round $t$, and we denote by $\mathbb{E}_{t,k} [\cdot] := \mathbb{E} [\cdot | \mathcal{P}_{t,k}]$, the conditional expectation given $\mathcal{P}_{t,k}$.

## A.2 SKELETON OF THE PROOF FOR THEOREM 1

Our convergence analysis follows the standard analysis of FedAvg (e.g., (Wang et al., 2020; Jhunjhunwala et al., 2022; Karimireddy et al., 2020)) considering additional error terms that are introduced by the presence of Byzantine clients. Let us first show that given that the number of sampled clients is sufficiently large, with a high probability for all $t \in [T]$, the number of Byzantine clients is bounded by $\hat{b}$. The proof is deferred to Appendix B.

**Lemma 1.** *Let $p < 1$ and $b$ be such that $0 < b/n < 1/2$. Consider* FedRo *as defined in Algorithm 1. Suppose that $\hat{n}$ and $\hat{b}$ are such that $b/n < \hat{b}/\hat{n} < 1/2$ and*

$$\hat{n} \geq \min \left\{ n, D \left( \frac{\hat{b}}{\hat{n}}, \frac{b}{n} \right)^{-1} \ln \left( \frac{T}{1-p} \right) \right\}, \tag{5}$$

*with $D(\alpha, \beta) := \alpha \ln (\alpha/\beta) + (1-\alpha) \ln (1-\alpha/1-\beta)$, for $\alpha, \beta \in (0, 1)$. Then, Event $\mathcal{E}$ as defined in (4) holds true with probability at least $p$.*

Now note that combining the global step of Algorithm 1, and (13), we have

$$x_{t+1} = x_t + \gamma_s \mathcal{A} \left( u_t^{(i)}, i \in \mathcal{S}_t \right) = x_t + \gamma_s \frac{1}{\hat{h}} \sum_{i \in \hat{\mathcal{H}}_t} u_t^{(i)} + \gamma_s e_t = x_t - K \gamma_s \gamma_c \frac{1}{\hat{h}} \sum_{i \in \hat{\mathcal{H}}_t} G_t^{(i)} + \gamma_s e_t.$$

Thus, denoting $\gamma := K \gamma_s \gamma_c$, we have

$$x_{t+1} = x_t - \gamma \frac{1}{\hat{h}} \sum_{i \in \hat{\mathcal{H}}_t} G_t^{(i)} + \gamma_s e_t, \tag{14}$$

Accordingly, the computation of $x_{t+1}$ is similar to FedAvg update but with an additional error term $\gamma_s e_t$. To bound this error, we first prove that the variance of the local updates from the honest clients is bounded as follows.

**Lemma 5.** *Consider Algorithm 1. Suppose that assumptions 1, 2, and 3 hold true. Suppose also that $\gamma_c \leq 1/16LK$ and $\gamma_c \gamma_s \leq 1/36LK$. Then for any $t \in \{0, \ldots, T-1\}$, and $K \geq 1$, we have*

$$\mathbb{E} \left[ \frac{1}{\hat{h}} \sum_{i \in \hat{\mathcal{H}}_t} \left\| x_{t,K}^{(i)} - \hat{x}_{t,K} \right\|^2 \right] \leq 3K\sigma^2 \gamma_c^2 + 18K^2 \gamma_c^2 \zeta^2,$$

*where $\hat{x}_{t,K} := \frac{1}{\hat{h}} \sum_{i \in \hat{\mathcal{H}}_t} x_{t,K}^{(i)}$.*

Combining the above lemma with the fact that the aggregation rule satisfies $(\hat{n}, \hat{b}, \kappa)$-robustness (Defintion 2), we can find a uniform bound on the expected error as follows.

**Lemma 6.** *Consider Algorithm 1. Suppose that assumptions 1, 2, and 3 hold true. Suppose also that $\gamma_c \leq 1/16LK$ and $\gamma_c \gamma_s \leq 1/36LK$ and the aggregation rule $\mathcal{A}$ is $(\hat{n}, \hat{b}, \kappa)$-robust. Recall the definition of $e_t$ from (13). Then for any $t \in \{0, \ldots, T-1\}$, and $K \geq 1$, we have*

$$\mathbb{E} \left[ \|e_t\|^2 \right] \leq 3K\kappa\sigma^2 \gamma_c^2 + 18K^2 \kappa \gamma_c^2 \zeta^2$$

Next, for any global step $t \in \{0, \ldots, T-1\}$, we analyze the change in the expected loss value i.e., $\mathbb{E} [F(x_{t+1})] - \mathbb{E} [F(x_t)]$. Formally, denoting by $h := |\mathcal{H}|$, the total number of honest clients and assuming[5] $h \geq 2$, we have the following lemma.

---

[5]When $h = 1$, as $b < n/2$, we must have $b = 0$ (Liu et al., 2021). Thus, there is only one client in the system. Assumption $h \geq 2$ is only made to avoid this trivial case, in the following results.

**Lemma 7.** *Consider Algorithm 1. Suppose that assumptions 1, 2, and 3 hold true. Suppose also that* $\gamma_c \leq 1/16LK$ *and* $\gamma_c\gamma_s \leq 1/36LK$ *and the aggregation rule* $\mathcal{A}$ *is* $(\hat{n}, \hat{b}, \kappa)$*-robust. Recall the definition of* $e_t$ *from (13). Then for any* $t \in \{0, \ldots, T-1\}$, *and* $K \geq 1$, *we have*

$$\mathbb{E}\left[F(x_{t+1})\right] \leq \mathbb{E}\left[F(x_t)\right] + \left(-\frac{2\gamma}{5} + 6L\gamma^2 + 8\gamma_c^2 L^2 K(K-1)\left(\frac{\gamma}{2} + 6L\gamma^2\right)\right)\mathbb{E}\left[\|\nabla F(x_t)\|^2\right]$$

$$+ 2L\gamma^2 \frac{\sigma^2}{\hat{h}K} + 6L\gamma^2 \frac{h-\hat{h}}{(h-1)\hat{h}}\zeta^2 + \left(\frac{\gamma}{2} + 6L\gamma^2\right)\left(2\gamma_c^2 L^2(K-1)\sigma^2 + 8\gamma_c^2 L^2 K(K-1)\zeta^2\right)$$

$$+ \left(\frac{10}{\gamma}\gamma_s^2 + L\gamma_s^2\right)\mathbb{E}\left[\|e_t\|^2\right].$$

### A.3 COMBINING ALL (PROOF OF THEOREM 1)

By Lemma 7, we have

$$\mathbb{E}\left[F(x_{t+1})\right] \leq \mathbb{E}\left[F(x_t)\right] + A\,\mathbb{E}\left[\|\nabla F(x_t)\|^2\right] + B\,\mathbb{E}\left[\|e_t\|^2\right].$$

$$+ 2L\gamma^2 \frac{\sigma^2}{\hat{h}K} + 6L\gamma^2 \frac{h-\hat{h}}{(h-1)\hat{h}}\zeta^2 + C\left(2\gamma_c^2 L^2(K-1)\sigma^2 + 8\gamma_c^2 L^2 K(K-1)\zeta^2\right),$$

where

$$A := -\frac{2\gamma}{5} + 6L\gamma^2 + 8\gamma_c^2 L^2 K(K-1)\left(\frac{\gamma}{2} + 6L\gamma^2\right),$$

$$B := \frac{\gamma}{2} + 6L\gamma^2$$

$$C := \frac{10}{\gamma}\gamma_s^2 + L\gamma_s^2$$

We now analyze $A$, $B$, and $C$ given that $\gamma \leq \frac{1}{36L}$, and $\gamma_c \leq \frac{1}{16KL}$. Then, we have

$$A \leq -\frac{2\gamma}{5} + \frac{\gamma}{6} + \frac{1}{32}\left(\frac{\gamma}{2} + 6L\gamma^2\right) \leq -\frac{\gamma}{5}$$

and

$$B \leq \frac{\gamma}{2} + \frac{\gamma}{6} \leq \frac{2\gamma}{3}$$

and

$$C \leq \frac{10}{\gamma}\gamma_s^2 + \frac{1}{36\gamma}\gamma_s^2 \leq \frac{11}{\gamma}\gamma_s^2$$

Therefore, we have

$$\mathbb{E}\left[F(x_{t+1})\right] \leq \mathbb{E}\left[F(x_t)\right] + \left(-\frac{\gamma}{5}\right)\mathbb{E}\left[\|\nabla F(x_t)\|^2\right] + \left(\frac{11}{\gamma}\gamma_s^2\right)\mathbb{E}\left[\|e_t\|^2\right]$$

$$+ 2L\gamma^2 \frac{\sigma^2}{\hat{h}K} + 6L\gamma^2 \frac{h-\hat{h}}{(h-1)\hat{h}}\zeta^2 + \left(\frac{2\gamma}{3}\right)\left(2\gamma_c^2 L^2(K-1)\sigma^2 + 8\gamma_c^2 L^2 K(K-1)\zeta^2\right).$$

Rearranging the terms, we obtain that

$$\mathbb{E}\left[\|\nabla F(x_t)\|^2\right] \leq \frac{5}{\gamma}\left(\mathbb{E}\left[F(x_t)\right] - \mathbb{E}\left[F(x_{t+1})\right]\right) + \left(\frac{55}{\gamma^2}\gamma_s^2\right)\mathbb{E}\left[\|e_t\|^2\right]$$

$$+ 10L\gamma \frac{\sigma^2}{\hat{h}K} + 30L\gamma \frac{h-\hat{h}}{(h-1)\hat{h}}\zeta^2 + \left(\frac{5}{3}\right)\left(2\gamma_c^2 L^2(K-1)\sigma^2 + 8\gamma_c^2 L^2 K(K-1)\zeta^2\right).$$

Then by Lemma 6, we have

$$\mathbb{E}\left[\|\nabla F(x_t)\|^2\right] \leq \frac{5}{\gamma}\left(\mathbb{E}\left[F(x_t)\right] - \mathbb{E}\left[F(x_{t+1})\right]\right) + \left(\frac{55}{\gamma^2}\gamma_s^2\right)\left(3K\kappa\sigma^2\gamma_c^2 + 18K^2\kappa\gamma_c^2\zeta^2\right)$$

$$+ 10L\gamma \frac{\sigma^2}{\hat{h}K} + 30L\gamma \frac{h-\hat{h}}{(h-1)\hat{h}}\zeta^2 + \left(\frac{5}{3}\right)\left(2\gamma_c^2 L^2(K-1)\sigma^2 + 8\gamma_c^2 L^2 K(K-1)\zeta^2\right).$$

Recall that $\gamma = K\gamma_c\gamma_s$, hence, we obtain that

$$
\begin{aligned}
\mathbb{E}\left[\|\nabla F(x_t)\|^2\right] &\leq \frac{5}{K\gamma_c\gamma_s}\left(\mathbb{E}\left[F(x_t)\right] - \mathbb{E}\left[F(x_{t+1})\right]\right) + 165\kappa\left(\frac{\sigma^2}{K} + 6\zeta^2\right) \\
&+ \frac{10LK\gamma_c\gamma_s}{\hat{h}}\left(\frac{\sigma^2}{K} + 3\frac{h-\hat{h}}{h-1}\zeta^2\right) + \frac{10}{3}\gamma_c^2 L^2(K-1)\sigma^2 + \frac{40}{3}\gamma_c^2 L^2 K(K-1)\zeta^2.
\end{aligned}
$$

Taking the average over $t \in \{0, \dots, T-1\}$, we have

$$
\begin{aligned}
\frac{1}{T}\sum_{t=0}^{T-1}\mathbb{E}\left[\|\nabla F(x_t)\|^2\right] &\leq \frac{5}{TK\gamma_c\gamma_s}\left(F(x_0) - \mathbb{E}\left[F(x_T)\right]\right) + 165\kappa\left(\frac{\sigma^2}{K} + 6\zeta^2\right) \\
&+ \frac{10LK\gamma_c\gamma_s}{\hat{h}}\left(\frac{\sigma^2}{K} + 3\frac{h-\hat{h}}{h-1}\zeta^2\right) + \frac{10}{3}\gamma_c^2 L^2(K-1)\sigma^2 + \frac{40}{3}\gamma_c^2 L^2 K(K-1)\zeta^2.
\end{aligned}
$$

Denoting $F^* = \min_{x\in\mathbb{R}^d} F(x)$, and noting that $\hat{h} \geq \hat{n}/2$, $h-1 \geq h/2$, we obtain that

$$
\begin{aligned}
\frac{1}{T}\sum_{t=0}^{T-1}\mathbb{E}\left[\|\nabla F(x_t)\|^2\right] &\leq \frac{5}{TK\gamma_c\gamma_s}\left(F(x_0) - F^*\right) + 165\kappa\left(\frac{\sigma^2}{K} + 6\zeta^2\right) \\
&+ \frac{20LK\gamma_c\gamma_s}{\hat{n}}\left(\frac{\sigma^2}{K} + 6\left(1 - \frac{\hat{h}}{h}\right)\zeta^2\right) + \frac{10}{3}\gamma_c^2 L^2(K-1)\sigma^2 + \frac{40}{3}\gamma_c^2 L^2 K(K-1)\zeta^2,
\end{aligned}
$$

Noting that $F(x_0) - F^* \leq \Delta_0$ yields the desired result.

### A.4 PROOF OF COROLLARY 1

*Proof.* We set

$$\gamma_s = 1, \tag{15}$$

and

$$
\gamma_c = \min\left\{\frac{1}{36LK}, \frac{1}{2K}\sqrt{\frac{\hat{n}\Delta_0}{LT\left(\frac{\sigma^2}{K} + 6\left(1 - \frac{\hat{h}}{h}\right)\zeta^2\right)}}, \sqrt[3]{\frac{3\Delta_0}{2K(K-1)TL^2(\sigma^2 + 4K\zeta^2)}}\right\}. \tag{16}
$$

Therefore, we have $\gamma_c \leq \frac{1}{16LK}$ and $\gamma_c\gamma_s \leq \frac{1}{36LK}$. Also, as $\frac{1}{\min\{a,b,c\}} = \max\{\frac{1}{a}, \frac{1}{b}, \frac{1}{c}\}$, for any $a, b > 0$, we obtain that

$$
\begin{aligned}
\frac{1}{\gamma_c} &= \max\left\{36LK, 2K\sqrt{\frac{LT\left(\frac{\sigma^2}{K} + 6\left(1 - \frac{\hat{h}}{h}\right)\zeta^2\right)}{\hat{n}\Delta_0}}, \sqrt[3]{\frac{2K(K-1)L^2T(\sigma^2 + 4K\zeta^2)}{3\Delta_0}}\right\} \\
&\leq 36LK + 2K\sqrt{\frac{LT\left(\frac{\sigma^2}{K} + 6\left(1 - \frac{\hat{h}}{h}\right)\zeta^2\right)}{\hat{n}\Delta_0}} + \sqrt[3]{\frac{2K(K-1)L^2T(\sigma^2 + 4K\zeta^2)}{3\Delta_0}}. \tag{17}
\end{aligned}
$$

Plugging (15), (16), and (17) into Theorem 1, we obtain that

$$
\frac{1}{T}\sum_{t=0}^{T-1}\mathbb{E}\left[\|\nabla F(x_t)\|^2\right] \overset{(a)}{\leq} \frac{180LK + 10K\sqrt{\frac{LT\left(\frac{\sigma^2}{K}+6\left(1-\frac{\hat{h}}{h}\right)\zeta^2\right)}{\hat{n}\Delta_0}} + \sqrt[3]{\frac{2K(K-1)L^2T(\sigma^2+4K\zeta^2)}{3\Delta_0}}}{TK}\Delta_0
$$

$$
+ \frac{20LK\gamma_c}{\hat{n}}\left(\frac{\sigma^2}{K}+6\left(1-\frac{\hat{h}}{h}\right)\zeta^2\right) + \frac{10}{3}\gamma_c^2 L^2(K-1)\sigma^2 + \frac{40}{3}\gamma_c^2 L^2 K(K-1)\zeta^2 + 165\kappa\left(\frac{\sigma^2}{K}+6\zeta^2\right)
$$

$$
\overset{(b)}{\leq} \frac{180L + 10\sqrt{\frac{LT\left(\frac{\sigma^2}{K}+6\left(1-\frac{\hat{h}}{h}\right)\zeta^2\right)}{\hat{n}\Delta_0}}}{T}\Delta_0 + \sqrt[3]{\frac{2(K-1)\Delta_0^2 L^2(\sigma^2+4K\zeta^2)}{3T^2K^2}} + 165\kappa\left(\frac{\sigma^2}{K}+6\zeta^2\right)
$$

$$
+ \frac{20LK\left(\frac{1}{2K}\sqrt{\frac{\hat{n}\Delta_0}{LT\left(\frac{\sigma^2}{K}+6\left(1-\frac{\hat{h}}{h}\right)\zeta^2\right)}}\right)}{\hat{n}}\left(\frac{\sigma^2}{K}+6\left(1-\frac{\hat{h}}{h}\right)\zeta^2\right)
$$

$$
+ \frac{10}{3}\left(\sqrt[3]{\frac{3\Delta_0}{2K(K-1)TL^2(\sigma^2+4K\zeta^2)}}\right)^2 L^2(K-1)(\sigma^2+4K\zeta^2),
$$

where (a) uses (17), and (b) uses the second and third terms of the minimum in (16). Rearranging the terms, we obtain that

$$
\frac{1}{T}\sum_{t=0}^{T-1}\mathbb{E}\left[\|\nabla F(x_t)\|^2\right] \leq 20\sqrt{\frac{L\Delta_0\left(\frac{\sigma^2}{K}+6\left(1-\frac{\hat{h}}{h}\right)\zeta^2\right)}{\hat{n}T}} + 165\kappa\left(\frac{\sigma^2}{K}+6\zeta^2\right)
$$

$$
+ \sqrt[3]{\frac{8(K-1)\Delta_0^2 L^2(\frac{\sigma^2}{K}+4\zeta^2)}{3KT^2}} + \frac{180L\Delta_0}{T}.
$$

Recalling $\hat{h}=\hat{n}-\hat{b}$, and $h=n-b$, and denoting $r=1-\frac{\hat{n}-\hat{b}}{n-b}$, we then have

$$
\frac{1}{T}\sum_{t=0}^{T-1}\mathbb{E}\left[\|\nabla F(x_t)\|^2\right] \in
$$

$$
\mathcal{O}\left(\sqrt{\frac{L\Delta_0}{\hat{n}T}\left(\frac{\sigma^2}{K}+r\zeta^2\right)} + \sqrt[3]{\frac{(1-\frac{1}{K})L^2\Delta_0^2(\frac{\sigma^2}{K}+\zeta^2)}{T^2}} + \frac{L\Delta_0}{T} + \kappa\left(\frac{\sigma^2}{K}+\zeta^2\right)\right),
$$

which is the desired result.

$\square$

### A.5 PROOF OF THE LEMMAS

First, we prove a useful lemma.

**Lemma 8.** *For any set $\{x^{(i)}\}_{i\in S}$ of $|S|$ vectors, we have*

$$
\frac{1}{|S|}\sum_{i\in S}\left\|x^{(i)}-\bar{x}\right\|^2 = \frac{1}{2}\cdot\frac{1}{|S|^2}\sum_{i,j\in S}\left\|x^{(i)}-x^{(j)}\right\|^2.
$$

*Proof.*

$$
\frac{1}{|S|^2} \sum_{i,j \in S} \left\| x^{(i)} - x^{(j)} \right\|^2 = \frac{1}{|S|^2} \sum_{i,j \in S} \left\| (x^{(i)} - \bar{x}) - (x^{(j)} - \bar{x}) \right\|^2
$$

$$
= \frac{1}{|S|^2} \sum_{i,j \in S} \left[ \left\| x^{(i)} - \bar{x} \right\|^2 + \left\| x^{(j)} - \bar{x} \right\|^2 + 2 \left\langle x^{(i)} - \bar{x}, \, x^{(j)} - \bar{x} \right\rangle \right]
$$

$$
= \frac{2}{|S|} \sum_{i,j \in S} \left\| x^{(i)} - \bar{x} \right\|^2 + \frac{2}{|S|^2} \sum_{i \in S} \left\langle x^{(i)} - \bar{x}, \sum_{j \in S} (x^{(j)} - \bar{x}) \right\rangle .
$$

Now as $\sum_{j \in S} (x^{(j)} - \bar{x}) = 0$, we conclude the desired result:

$$
\frac{1}{|S|^2} \sum_{i,j \in S} \left\| x^{(i)} - x^{(j)} \right\|^2 = \frac{2}{|S|} \sum_{i,j \in S} \left\| x^{(i)} - \bar{x} \right\|^2 .
$$

$\square$

**Lemma 5.** *Consider Algorithm 1. Suppose that assumptions 1, 2, and 3 hold true. Suppose also that $\gamma_c \leq 1/16LK$ and $\gamma_c \gamma_s \leq 1/36LK$. Then for any $t \in \{0, \ldots, T-1\}$, and $K \geq 1$, we have*

$$
\mathbb{E} \left[ \frac{1}{\hat{h}} \sum_{i \in \hat{\mathcal{H}}_t} \left\| x_{t,K}^{(i)} - \hat{x}_{t,K} \right\|^2 \right] \leq 3K\sigma^2 \gamma_c^2 + 18K^2 \gamma_c^2 \zeta^2,
$$

*where $\hat{x}_{t,K} := \frac{1}{\hat{h}} \sum_{i \in \hat{\mathcal{H}}_t} x_{t,K}^{(i)}$.*

*Proof.* For any $k \in [K]$, we have

$$
x_{t,k}^{(i)} - x_{t,k}^{(j)} = x_{t,k-1}^{(i)} - \gamma_c g_{t,k-1}^{(i)} - x_{t,k-1}^{(j)} + \gamma_c g_{t,k-1}^{(j)}
$$

$$
= x_{t,k-1}^{(i)} - \gamma_c \left( g_{t,k-1}^{(i)} - \nabla f^{(i)}(x_{t,k-1}^{(i)}) \right) - \gamma_c \left( \nabla f^{(i)}(x_{t,k-1}^{(i)}) - \nabla f^{(i)}(\hat{x}_{t,k-1}) \right)
$$

$$
- \gamma_c \left( \nabla f^{(i)}(\hat{x}_{t,k-1}) - \nabla f^{(j)}(\hat{x}_{t,k-1}) \right)
$$

$$
- \gamma_c \left( \nabla f^{(j)}(\hat{x}_{t,k-1}) - \nabla f^{(j)}(x_{t,k-1}^{(j)}) \right) - \gamma_c \left( \nabla f^{(j)}(x_{t,k-1}^{(j)}) - g_{t,k-1}^{(j)} \right) - x_{t,k-1}^{(j)}
$$

Now denoting

$$
D := -\gamma_c \left( \nabla f^{(i)}(x_{t,k-1}^{(i)}) - \nabla f^{(i)}(\hat{x}_{t,k-1}) \right) - \gamma_c \left( \nabla f^{(i)}(\hat{x}_{t,k-1}) - \nabla f^{(j)}(\hat{x}_{t,k-1}) \right)
$$

$$
- \gamma_c \left( \nabla f^{(j)}(\hat{x}_{t,k-1}) - \nabla f^{(j)}(x_{t,k-1}^{(j)}) \right),
$$

we have

$$
x_{t,k}^{(i)} - x_{t,k}^{(j)} = x_{t,k-1}^{(i)} - \gamma_c \left( g_{t,k-1}^{(i)} - \nabla f^{(i)}(x_{t,k-1}^{(i)}) \right) + D - \gamma_c \left( \nabla f^{(j)}(x_{t,k-1}^{(j)}) - g_{t,k-1}^{(j)} \right) - x_{t,k-1}^{(j)}
$$

$$
= x_{t,k-1}^{(i)} - x_{t,k-1}^{(j)} + D - \gamma_c \left( g_{t,k-1}^{(i)} - \nabla f^{(i)}(x_{t,k-1}^{(i)}) - \nabla f^{(j)}(x_{t,k-1}^{(j)}) - g_{t,k-1}^{(j)} \right).
$$

Now note that given the history $\mathcal{P}_{t,k-1}$, $D$ is a constant. Also, $\mathbb{E}_{t,k-1} \left[ g_{t,k-1}^{(i)} - \nabla f^{(i)}(x_{t,k-1}^{(i)}) \right] = 0$ and $\mathbb{E}_{t,k-1} \left[ g_{t,k-1}^{(j)} - \nabla f^{(j)}(x_{t,k-1}^{(j)}) \right] = 0$. Therefore,

$$
\mathbb{E}_{t,k-1} \left[ \left\| x_{t,k}^{(i)} - x_{t,k}^{(j)} \right\|^2 \right] \leq 2\sigma^2 \gamma_c^2 + \left\| x_{t,k-1}^{(i)} - x_{t,k-1}^{(j)} + D \right\|^2
$$

by Young's inequality for any $\alpha > 0$, we have

$$
\mathbb{E}_{t,k-1} \left[ \left\| x_{t,k}^{(i)} - x_{t,k}^{(j)} \right\|^2 \right] \leq 2\sigma^2 \gamma_c^2 + (1 + \frac{1}{\alpha}) \left\| x_{t,k-1}^{(i)} - x_{t,k-1}^{(j)} \right\|^2 + (1 + \alpha) \left\| D \right\|^2 .
$$

Now by the definition of $D$ and using the fact that (by Jensen's inequality) for $a, b, c \geq 0$, $(a + b + c)^2 \leq 3a^2 + 3b^2 + 3c^2$, we have

$$
\begin{aligned}
\mathbb{E}_{t,k-1}\left[\left\|x_{t,k}^{(i)} - x_{t,k}^{(j)}\right\|^2\right] &\leq 2\sigma^2\gamma_c^2 + (1 + \frac{1}{\alpha})\left\|x_{t,k-1}^{(i)} - x_{t,k-1}^{(j)}\right\|^2 \\
&+ 3(1+\alpha)\gamma_c^2\left\|\nabla f^{(i)}(x_{t,k-1}^{(i)}) - \nabla f^{(i)}(\hat{x}_{t,k-1})\right\|^2 \\
&+ 3(1+\alpha)\gamma_c^2\left\|\nabla f^{(i)}(\hat{x}_{t,k-1}) - \nabla f^{(j)}(\hat{x}_{t,k-1})\right\|^2 \\
&+ 3(1+\alpha)\gamma_c^2\left\|\nabla f^{(j)}(x_{t,k-1}^{(j)}) - \nabla f^{(j)}(\hat{x}_{t,k-1})\right\|^2.
\end{aligned}
$$

Taking the average over $i, j \in \hat{\mathcal{H}}_t$, we have

$$
\begin{aligned}
\frac{1}{\hat{h}^2}\sum_{i,j\in\hat{\mathcal{H}}_t}\mathbb{E}_{t,k-1}\left[\left\|x_{t,k}^{(i)} - x_{t,k}^{(j)}\right\|^2\right] &\leq 2\sigma^2\gamma_c^2 + \frac{1}{\hat{h}^2}\sum_{i,j\in\hat{\mathcal{H}}_t}(1 + \frac{1}{\alpha})\left\|x_{t,k-1}^{(i)} - x_{t,k-1}^{(j)}\right\|^2 \qquad (18)\\
&+ 6(1+\alpha)\gamma_c^2\frac{1}{\hat{h}}\sum_{i\in\hat{\mathcal{H}}_t}\left\|\nabla f^{(i)}(x_{t,k-1}^{(i)}) - \nabla f^{(i)}(\hat{x}_{t,k-1})\right\|^2 \\
&+ 3(1+\alpha)\gamma_c^2\frac{1}{\hat{h}^2}\sum_{i,j\in\hat{\mathcal{H}}_t}\left\|\nabla f^{(i)}(\hat{x}_{t,k-1}) - \nabla f^{(j)}(\hat{x}_{t,k-1})\right\|^2
\end{aligned}
$$

Now note that by Assumption 1, we have

$$
\begin{aligned}
\frac{1}{\hat{h}}\sum_{i\in\hat{\mathcal{H}}_t}\left\|\nabla f^{(i)}(x_{t,k-1}^{(i)}) - \nabla f^{(i)}(\hat{x}_{t,k-1})\right\|^2 &\leq L^2\frac{1}{\hat{h}}\sum_{i\in\hat{\mathcal{H}}_t}\left\|x_{t,k-1}^{(i)} - \hat{x}_{t,k-1}\right\|^2 \\
&\leq \frac{L^2}{\hat{h}^2}\sum_{i,j\in\hat{\mathcal{H}}_t}\left\|x_{t,k-1}^{(i)} - x_{t,k-1}^{(j)}\right\|^2 \qquad (19)
\end{aligned}
$$

where in the second inequality we used Lemma 8. Also, by Lemma 8, we have

$$
\frac{1}{\hat{h}^2}\sum_{i,j\in\hat{\mathcal{H}}_t}\left\|\nabla f^{(i)}(\hat{x}_{t,k-1}) - \nabla f^{(j)}(\hat{x}_{t,k-1})\right\|^2 = \frac{2}{\hat{h}}\sum_{i\in\hat{\mathcal{H}}_t}\left\|\nabla f^{(i)}(\hat{x}_{t,k-1}) - \nabla\hat{f}(\hat{x}_{t,k-1})\right\|^2, \quad (20)
$$

where $\hat{f}(x) = \frac{1}{\hat{h}}\sum_{i\in\hat{\mathcal{H}}_t}f^{(i)}(x)$. Plugging (19), and (20) into (18), we have

$$
\begin{aligned}
\frac{1}{\hat{h}^2}\sum_{i,j\in\hat{\mathcal{H}}_t}\mathbb{E}_{t,k-1}\left[\left\|x_{t,k}^{(i)} - x_{t,k}^{(j)}\right\|^2\right] &\leq 2\sigma^2\gamma_c^2 + 6(1+\alpha)\gamma_c^2\frac{1}{\hat{h}}\sum_{i\in\hat{\mathcal{H}}_t}\left\|\nabla f^{(i)}(\hat{x}_{t,k-1}) - \nabla\hat{f}(\hat{x}_{t,k-1})\right\|^2 \\
&+ (1 + \frac{1}{\alpha})\frac{1}{\hat{h}^2}\sum_{i,j\in\hat{\mathcal{H}}_t}\left\|x_{t,k-1}^{(i)} - x_{t,k-1}^{(j)}\right\|^2 \\
&+ 6(1+\alpha)L^2\gamma_c^2\frac{1}{\hat{h}^2}\sum_{i,j\in\hat{\mathcal{H}}_t}\left\|x_{t,k-1}^{(i)} - x_{t,k-1}^{(j)}\right\|^2.
\end{aligned}
$$

Taking the total expectation from both sides, we have

$$
\begin{aligned}
\mathbb{E}\left[\frac{1}{\hat{h}^2}\sum_{i,j\in\hat{\mathcal{H}}_t}\left\|x_{t,k}^{(i)} - x_{t,k}^{(j)}\right\|^2\right] &\leq 2\sigma^2\gamma_c^2 + 6(1+\alpha)\gamma_c^2\,\mathbb{E}\left[\frac{1}{\hat{h}}\sum_{i\in\hat{\mathcal{H}}_t}\left\|\nabla f^{(i)}(\hat{x}_{t,k-1}) - \nabla\hat{f}(\hat{x}_{t,k-1})\right\|^2\right] \\
&+ (1 + \frac{1}{\alpha})\,\mathbb{E}\left[\frac{1}{\hat{h}^2}\sum_{i,j\in\hat{\mathcal{H}}_t}\left\|x_{t,k-1}^{(i)} - x_{t,k-1}^{(j)}\right\|^2\right] \\
&+ 6(1+\alpha)L^2\gamma_c^2\,\mathbb{E}\left[\frac{1}{\hat{h}^2}\sum_{i,j\in\hat{\mathcal{H}}_t}\left\|x_{t,k-1}^{(i)} - x_{t,k-1}^{(j)}\right\|^2\right] \qquad (21)
\end{aligned}
$$

Now recall that $\hat{\mathcal{H}}_t \sim \mathcal{U}^{\hat{h}}(\mathcal{H})$, therefore, for any $x \in \mathbb{R}^d$, we have

$$\mathbb{E}\left[\frac{1}{\hat{h}} \sum_{i \in \hat{\mathcal{H}}_t} \left\|\nabla f^{(i)}(x) - \nabla F(x)\right\|^2\right] = \frac{1}{|\mathcal{H}|} \sum_{i \in \mathcal{H}} \left\|\nabla f^{(i)}(x) - \nabla F(x)\right\|^2.$$

Also, as $\nabla \hat{f}(x)$ is the minimizer of the function $g(z) := \frac{1}{\hat{h}} \sum_{i \in \hat{\mathcal{H}}_t} \left\|\nabla f^{(i)}(x) - z\right\|^2$, for any $x \in \mathbb{R}^d$, we have

$$\mathbb{E}\left[\frac{1}{\hat{h}} \sum_{i \in \hat{\mathcal{H}}_t} \left\|\nabla f^{(i)}(x) - \nabla \hat{f}(x)\right\|^2\right] \leq \mathbb{E}\left[\frac{1}{\hat{h}} \sum_{i \in \hat{\mathcal{H}}_t} \left\|\nabla f^{(i)}(x) - \nabla F(x)\right\|^2\right]$$

$$\leq \frac{1}{|\mathcal{H}|} \sum_{i \in \mathcal{H}} \left\|\nabla f^{(i)}(x) - \nabla F(x)\right\|^2$$

$$\leq \zeta^2,$$

where in the last inequality we used Assumption 3. Combining this with (21), and denoting $u_k := \mathbb{E}\left[\frac{1}{\hat{h}^2} \sum_{i,j \in \hat{\mathcal{H}}_t} \left\|x_{t,k}^{(i)} - x_{t,k}^{(j)}\right\|^2\right]$, we obtain that

$$u_k \leq (1 + \frac{1}{\alpha} + 6(1+\alpha)L^2\gamma_c^2)u_{k-1} + 2\sigma^2\gamma_c^2 + 6(1+\alpha)\gamma_c^2\zeta^2.$$

For $\alpha = 2K - 1$, we have

$$u_k \leq (1 + \frac{1}{2K-1} + 12KL^2\gamma_c^2)u_{k-1} + 2\sigma^2\gamma_c^2 + 12K\gamma_c^2\zeta^2$$

As $\gamma_c^2 \leq \frac{1}{24K^2L^2}$, we have[6]

$$u_k \leq (1 + \frac{1}{K-1})u_{k-1} + 2\sigma^2\gamma_c^2 + 12K\gamma_c^2\zeta^2.$$

Therefore,

$$u_K \leq \left(2\sigma^2\gamma_c^2 + 12K\gamma_c^2\zeta^2\right) \sum_{\tau=0}^{K-1} \left(1 + \frac{1}{K-1}\right)^\tau$$

$$\leq \left(2\sigma^2\gamma_c^2 + 12K\gamma_c^2\zeta^2\right) K \left(1 + \frac{1}{K-1}\right)^{K-1}$$

$$\stackrel{(a)}{\leq} \left(2\sigma^2\gamma_c^2 + 12K\gamma_c^2\zeta^2\right) Ke$$

$$\leq 3K \left(2\sigma^2\gamma_c^2 + 12K\gamma_c^2\zeta^2\right)$$

$$= 6K\sigma^2\gamma_c^2 + 36K^2\gamma_c^2\zeta^2, \tag{22}$$

where $e$ is Euler's number and (a) uses the facts that $\left(1 + \frac{1}{K-1}\right)^{K-1}$ is increasing in K and

$$\lim_{K \to \infty} \left(1 + \frac{1}{K-1}\right)^{K-1} = e.$$

Now note that by Lemma 8, we have

$$\frac{1}{\hat{h}} \sum_{i \in \hat{\mathcal{H}}_t} \left\|x_{t,K}^{(i)} - \hat{x}_{t,K}\right\|^2 = \frac{1}{2\hat{h}^2} \sum_{i,j \in \hat{\mathcal{H}}_t} \left\|x_{t,K}^{(i)} - x_{t,K}^{(j)}\right\|^2.$$

Combining this with (22) yields the result.

$\square$

---

[6]Note that here the results holds trivially for $K = 1$, so hereafter we assume $K \geq 2$.

**Lemma 6.** *Consider Algorithm 1. Suppose that assumptions 1, 2, and 3 hold true. Suppose also that* $\gamma_c \leq 1/16LK$ *and* $\gamma_c \gamma_s \leq 1/36LK$ *and the aggregation rule* $\mathcal{A}$ *is* $(\hat{n}, \hat{b}, \kappa)$*-robust. Recall the definition of* $e_t$ *from (13). Then for any* $t \in \{0, \ldots, T-1\}$*, and* $K \geq 1$*, we have*

$$\mathbb{E}\left[\|e_t\|^2\right] \leq 3K\kappa\sigma^2\gamma_c^2 + 18K^2\kappa\gamma_c^2\zeta^2$$

*Proof.* Follows directly from Lemma 5 and the fact the aggregation rule $\mathcal{A}$ is $(\hat{n}, \hat{b}, \kappa)$-robust (Definition 2). □

Before proving Lemma 7, we prove a few useful lemmas.

**Lemma 9** (Equation (3) in (Yang et al., 2021)). *Consider Algorithm 1. Suppose that Assumption 2 holds true. Then for any* $t \in \{0, \ldots, T-1\}$*, and* $K \geq 1$*, we have*

$$\mathbb{E}_t\left[\left\|\frac{1}{\hat{h}}\sum_{i\in\hat{\mathcal{H}}_t} G_t^{(i)}\right\|^2\right] \leq \frac{2\sigma^2}{\hat{h}K} + 2\mathbb{E}_t\left[\left\|\frac{1}{\hat{h}}\sum_{i\in\hat{\mathcal{H}}_t} \Delta_t^{(i)}\right\|^2\right].$$

*Proof.*

$$\mathbb{E}_t\left[\left\|\frac{1}{\hat{h}}\sum_{i\in\hat{\mathcal{H}}_t} G_t^{(i)}\right\|^2\right] = \mathbb{E}_t\left[\left\|\frac{1}{\hat{h}}\sum_{i\in\hat{\mathcal{H}}_t}\frac{1}{K}\sum_{k=0}^{K-1} g_{t,k}^{(i)}\right\|^2\right]$$

$$= \mathbb{E}_t\left[\left\|\frac{1}{\hat{h}}\frac{1}{K}\sum_{i\in\hat{\mathcal{H}}_t}\sum_{k=0}^{K-1}(g_{t,k}^{(i)} - \nabla f^{(i)}(x_{t,k}^{(i)}) + \nabla f^{(i)}(x_{t,k}^{(i)}))\right\|^2\right]$$

$$\leq 2\mathbb{E}_t\left[\left\|\frac{1}{\hat{h}}\frac{1}{K}\sum_{i\in\hat{\mathcal{H}}_t}\sum_{k=0}^{K-1}(g_{t,k}^{(i)} - \nabla f^{(i)}(x_{t,k}^{(i)}))\right\|^2\right] + 2\mathbb{E}_t\left[\left\|\frac{1}{\hat{h}}\frac{1}{K}\sum_{i\in\hat{\mathcal{H}}_t}\sum_{k=0}^{K-1}\nabla f^{(i)}(x_{t,k}^{(i)})\right\|^2\right],$$

where we used Young's inequality. Now note that by definition, we have

$$\frac{1}{\hat{h}}\frac{1}{K}\sum_{i\in\hat{\mathcal{H}}_t}\sum_{k=0}^{K-1}\nabla f^{(i)}(x_{t,k}^{(i)}) = \frac{1}{\hat{h}}\sum_{i\in\hat{\mathcal{H}}_t}\Delta_t^{(i)}.$$

Also,

$$\mathbb{E}_t\left[\left\|\frac{1}{\hat{h}}\frac{1}{K}\sum_{i\in\hat{\mathcal{H}}_t}\sum_{k=0}^{K-1}(g_{t,k}^{(i)} - \nabla f^{(i)}(x_{t,k}^{(i)}))\right\|^2\right] = \frac{1}{\hat{h}^2 K^2}\mathbb{E}_t\left[\left\|\sum_{k=0}^{K-1}\sum_{i\in\hat{\mathcal{H}}_t}(g_{t,k}^{(i)} - \nabla f^{(i)}(x_{t,k}^{(i)}))\right\|^2\right]$$

$$= \frac{1}{\hat{h}^2 K^2}\mathbb{E}_t\left[\mathbb{E}_{t,K-1}\left[\left\|\sum_{i\in\hat{\mathcal{H}}_t}(g_{t,K-1}^{(i)} - \nabla f^{(i)}(x_{t,K-1}^{(i)})) + \sum_{k=0}^{K-2}\sum_{i\in\hat{\mathcal{H}}_t}(g_{t,k}^{(i)} - \nabla f^{(i)}(x_{t,k}^{(i)}))\right\|^2\right]\right].$$

Now note that given the history $\mathcal{P}_{t,K-1}$, the second term of above is constant and by Assumption 2, for each $i \in \hat{\mathcal{H}}_t$, we have

$$\mathbb{E}_{t,K-1}\left[g_{t,K-1}^{(i)}\right] = \nabla f^{(i)}(x_{t,K-1}^{(i)}) \quad \text{and} \quad \mathbb{E}_{t,K-1}\left[\left\|g_{t,K-1}^{(i)} - \nabla f^{(i)}(x_{t,K-1}^{(i)})\right\|^2\right] \leq \sigma^2.$$

Therefore, as the stochastic gradients computed on different honest clients are independent of each other, we have

$$\mathbb{E}_t\left[\left\|\frac{1}{\hat{h}}\frac{1}{K}\sum_{i\in\hat{\mathcal{H}}_t}\sum_{k=0}^{K-1}(g_{t,k}^{(i)} - \nabla f^{(i)}(x_{t,k}^{(i)}))\right\|^2\right] \leq \frac{1}{\hat{h}^2 K^2}\left(\mathbb{E}_t\left[\left\|\sum_{k=0}^{K-2}\sum_{i\in\hat{\mathcal{H}}_t}(g_{t,k}^{(i)} - \nabla f^{(i)}(x_{t,k}^{(i)}))\right\|^2\right] + \hat{h}\sigma^2\right).$$

Using the same approach for $k = \{K - 2, \ldots, 0\}$, we obtain that

$$\mathbb{E}_t \left[ \left\| \frac{1}{\hat{h}} \frac{1}{K} \sum_{i \in \hat{\mathcal{H}}_t} \sum_{k=0}^{K-1} (g_{t,k}^{(i)} - \nabla f^{(i)}(x_{t,k}^{(i)})) \right\|^2 \right] \leq \frac{\sigma^2}{K\hat{h}}.$$

This completes the proof. $\qquad\square$

**Lemma 10.** *Consider Algorithm 1. Suppose that assumptions 1, and 2 hold true. Recall that $\gamma := K\gamma_s\gamma_c$. Recall also the definition of $e_t$ from (13). Then for any $t \in \{0, \ldots, T-1\}$, and $K \geq 1$, we have*

$$\mathbb{E}_t \left[ F(x_{t+1}) \right] \leq F(x_t) - \frac{\gamma}{2} \left( \|\nabla F(x_t)\|^2 + \mathbb{E}_t \left[ \left\| \frac{1}{h} \sum_{i \in \mathcal{H}} \Delta_t^{(i)} \right\|^2 \right] - \frac{1}{h} \sum_{i \in \mathcal{H}} \mathbb{E}_t \left[ \left\| \nabla f^{(i)}(x_t) - \Delta_t^{(i)} \right\|^2 \right] \right)$$

$$+ L\gamma^2 \left( \frac{2\sigma^2}{\hat{h}K} + 2\mathbb{E}_t \left[ \left\| \frac{1}{\hat{h}} \sum_{i \in \hat{\mathcal{H}}_t} \Delta_t^{(i)} \right\|^2 \right] \right) + \frac{\gamma}{10} \|\nabla F(x_t)\|^2 + \frac{10}{\gamma} \gamma_s^2 \mathbb{E}_t \left[ \|e_t\|^2 \right] + L\gamma_s^2 \mathbb{E}_t \left[ \|e_t\|^2 \right]$$

*Proof.* Using smoothness (Assumption 1), and (14), we have

$$\mathbb{E}_t \left[ F(x_{t+1}) \right] \leq F(x_t) + \left\langle \nabla F(x_t), \mathbb{E}_t \left[ x_{t+1} - x_t \right] \right\rangle + \frac{L}{2} \mathbb{E}_t \left[ \|x_{t+1} - x_t\|^2 \right]$$

$$= F(x_t) + \left\langle \nabla F(x_t), \mathbb{E}_t \left[ -\gamma \frac{1}{\hat{h}} \sum_{i \in \hat{\mathcal{H}}_t} G_t^{(i)} + \gamma_s e_t \right] \right\rangle + \frac{L}{2} \mathbb{E}_t \left[ \left\| -\gamma \frac{1}{\hat{h}} \sum_{i \in \hat{\mathcal{H}}_t} G_t^{(i)} + \gamma_s e_t \right\|^2 \right]$$

$$\overset{(a)}{\leq} F(x_t) - \gamma \mathbb{E}_t \left[ \left\langle \nabla F(x_t), \frac{1}{\hat{h}} \sum_{i \in \hat{\mathcal{H}}_t} G_t^{(i)} \right\rangle \right] + L\gamma^2 \mathbb{E}_t \left[ \left\| \frac{1}{\hat{h}} \sum_{i \in \hat{\mathcal{H}}_t} G_t^{(i)} \right\|^2 \right]$$

$$+ \mathbb{E}_t \left[ \left\langle \nabla F(x_t), \gamma_s e_t \right\rangle \right] + L\gamma_s^2 \mathbb{E}_t \left[ \|e_t\|^2 \right]$$

$$\overset{(b)}{\leq} F(x_t) - \gamma \mathbb{E}_t \left[ \left\langle \nabla F(x_t), \frac{1}{\hat{h}} \sum_{i \in \hat{\mathcal{H}}_t} G_t^{(i)} \right\rangle \right] + L\gamma^2 \mathbb{E}_t \left[ \left\| \frac{1}{\hat{h}} \sum_{i \in \hat{\mathcal{H}}_t} G_t^{(i)} \right\|^2 \right]$$

$$+ \frac{\gamma}{10} \|\nabla F(x_t)\|^2 + \frac{10}{\gamma} \gamma_s^2 \mathbb{E}_t \left[ \|e_t\|^2 \right] + L\gamma_s^2 \mathbb{E}_t \left[ \|e_t\|^2 \right],$$

where (a) uses Young's inequality and (b) uses the fact that $\langle a, b \rangle \leq \frac{\|a\|^2}{c} + c\|b\|^2$. We now have

$$- \gamma \mathbb{E}_t \left[ \left\langle \nabla F(x_t), \frac{1}{\hat{h}} \sum_{i \in \hat{\mathcal{H}}_t} G_t^{(i)} \right\rangle \right] = -\gamma \left\langle \nabla F(x_t), \mathbb{E}_{\hat{\mathcal{H}}_t \sim \mathcal{U}(\mathcal{H})} \left[ \mathbb{E}_t \left[ \frac{1}{\hat{h}} \sum_{i \in \hat{\mathcal{H}}_t} G_t^{(i)} | \hat{\mathcal{H}}_t \right] \right] \right\rangle$$

$$\overset{(a)}{=} -\gamma \left\langle \nabla F(x_t), \mathbb{E}_{\hat{\mathcal{H}}_t \sim \mathcal{U}(\mathcal{H})} \left[ \mathbb{E}_t \left[ \frac{1}{\hat{h}} \sum_{i \in \hat{\mathcal{H}}_t} \Delta_t^{(i)} | \hat{\mathcal{H}}_t \right] \right] \right\rangle$$

$$= -\gamma \mathbb{E}_t \left[ \left\langle \nabla F(x_t), \frac{1}{\hat{h}} \sum_{i \in \hat{\mathcal{H}}_t} \Delta_t^{(i)} \right\rangle \right]$$

$$= \frac{\gamma}{2} \left( \mathbb{E}_t \left[ \left\| \nabla F(x_t) - \frac{1}{h} \sum_{i \in \mathcal{H}} \Delta_t^{(i)} \right\|^2 \right] - \|\nabla F(x_t)\|^2 - \mathbb{E}_t \left[ \left\| \frac{1}{h} \sum_{i \in \mathcal{H}} \Delta_t^{(i)} \right\|^2 \right] \right),$$

where (a) comes from the fact that

$$\mathbb{E}_t \left[ g_{t,k}^{(i)} | i \in \hat{\mathcal{H}}_t \right] = \mathbb{E}_t \left[ \mathbb{E}_{t,k-1} \left[ g_{t,k}^{(i)} | i \in \hat{\mathcal{H}}_t \right] \right] = \mathbb{E}_t \left[ \mathbb{E}_{t,k-1} \left[ \nabla f^{(i)}(x_{t,k}^{(i)}) | i \in \hat{\mathcal{H}}_t \right] \right].$$

By Lemma 9, we also have

$$\mathbb{E}_t \left[ \left\| \frac{1}{\hat{h}} \sum_{i \in \hat{\mathcal{H}}_t} G_t^{(i)} \right\|^2 \right] \leq \frac{2\sigma^2}{\hat{h}K} + 2\mathbb{E}_t \left[ \left\| \frac{1}{\hat{h}} \sum_{i \in \hat{\mathcal{H}}_t} \Delta_t^{(i)} \right\|^2 \right].$$

Combining all we have

$$\mathbb{E}_t \left[ F(x_{t+1}) \right] \leq F(x_t) - \frac{\gamma}{2} \left( \|\nabla F(x_t)\|^2 + \mathbb{E}_t \left[ \left\| \frac{1}{h} \sum_{i \in \mathcal{H}} \Delta_t^{(i)} \right\|^2 \right] - \mathbb{E}_t \left[ \left\| \nabla F(x_t) - \frac{1}{h} \sum_{i \in \mathcal{H}} \Delta_t^{(i)} \right\|^2 \right] \right)$$

$$+ L\gamma^2 \left( \frac{2\sigma^2}{MK} + 2\mathbb{E}_t \left[ \left\| \frac{1}{\hat{h}} \sum_{i \in \hat{\mathcal{H}}_t} \Delta_t^{(i)} \right\|^2 \right] \right) + \frac{\gamma}{10} \|\nabla F(x_t)\|^2 + \frac{10}{\gamma} \gamma_s^2 \mathbb{E}_t \left[ \|e_t\|^2 \right] + L\gamma_s^2 \mathbb{E}_t \left[ \|e_t\|^2 \right]$$

$\square$

**Lemma 11** (Lemma 7 in (Jhunjhunwala et al., 2022)). *Consider Algorithm 1. Suppose that assumptions 1, 2, and 3 hold true. Then for any $t \in \{0, \ldots, T-1\}$, and $K \geq 1$, we have*

$$\mathbb{E}_t \left[ \left\| \frac{1}{\hat{h}} \sum_{i \in \hat{\mathcal{H}}_t} \Delta_t^{(i)} \right\|^2 \right] \leq \frac{3}{h} \sum_{i \in \mathcal{H}} \mathbb{E}_t \left[ \left\| \nabla f^{(i)}(x_t) - \Delta_t^{(i)} \right\|^2 \right] + \frac{3(h - \hat{h})}{(h-1)\hat{h}} \zeta^2 + 3 \|\nabla F(x_t)\|^2$$

**Lemma 12** (Lemma 6 in (Jhunjhunwala et al., 2022)). *Consider Algorithm 1. Suppose that assumptions 1, 2, and 3 hold true. Then for any $t \in \{0, \ldots, T-1\}$, and $K \geq 1$, we have*

$$\frac{1}{h} \sum_{i \in \mathcal{H}} \mathbb{E}_t \left[ \left\| \nabla f^{(i)}(x_t) - \Delta_t^{(i)} \right\|^2 \right] \leq 2\gamma_c^2 L^2 (K-1)\sigma^2 + 8\gamma_c^2 L^2 K(K-1) \left( \zeta^2 + \|\nabla F(x_t)\|^2 \right).$$

**Lemma 7.** *Consider Algorithm 1. Suppose that assumptions 1, 2, and 3 hold true. Suppose also that $\gamma_c \leq 1/16LK$ and $\gamma_c \gamma_s \leq 1/36LK$ and the aggregation rule $\mathcal{A}$ is $(\hat{n}, \hat{b}, \kappa)$-robust. Recall the definition of $e_t$ from (13). Then for any $t \in \{0, \ldots, T-1\}$, and $K \geq 1$, we have*

$$\mathbb{E}\left[ F(x_{t+1}) \right] \leq \mathbb{E}\left[ F(x_t) \right] + \left( -\frac{2\gamma}{5} + 6L\gamma^2 + 8\gamma_c^2 L^2 K(K-1) \left( \frac{\gamma}{2} + 6L\gamma^2 \right) \right) \mathbb{E}\left[ \|\nabla F(x_t)\|^2 \right]$$

$$+ 2L\gamma^2 \frac{\sigma^2}{\hat{h}K} + 6L\gamma^2 \frac{h - \hat{h}}{(h-1)\hat{h}} \zeta^2 + \left( \frac{\gamma}{2} + 6L\gamma^2 \right) \left( 2\gamma_c^2 L^2 (K-1)\sigma^2 + 8\gamma_c^2 L^2 K(K-1)\zeta^2 \right)$$

$$+ \left( \frac{10}{\gamma} \gamma_s^2 + L\gamma_s^2 \right) \mathbb{E}\left[ \|e_t\|^2 \right].$$

*Proof.* Combining lemmas 10 and 11, we obtain that

$$\mathbb{E}_t \left[ F(x_{t+1}) \right] \leq F(x_t) + \left( -\frac{\gamma}{2} + \frac{\gamma}{10} + 6L\gamma^2 \right) \|\nabla F(x_t)\|^2 + 2L\gamma^2 \frac{\sigma^2}{\hat{h}K} + 6L\gamma^2 \frac{h - \hat{h}}{(h-1)\hat{h}} \zeta^2$$

$$+ \left( \frac{\gamma}{2} + 6L\gamma^2 \right) \frac{1}{h} \sum_{i \in \mathcal{H}} \mathbb{E}_t \left[ \left\| \nabla f^{(i)}(x_t) - \Delta_t^{(i)} \right\|^2 \right] + \left( \frac{10}{\gamma} \gamma_s^2 + L\gamma_s^2 \right) \mathbb{E}_t \left[ \|e_t\|^2 \right]$$

Combining this with Lemma 12, we have

$$\mathbb{E}_t \left[ F(x_{t+1}) \right] \leq F(x_t) + \left( -\frac{2\gamma}{5} + 6L\gamma^2 + 8\gamma_c^2 L^2 K(K-1) \left( \frac{\gamma}{2} + 6L\gamma^2 \right) \right) \|\nabla F(x_t)\|^2$$

$$+ 2L\gamma^2 \frac{\sigma^2}{\hat{h}K} + 6L\gamma^2 \frac{h - \hat{h}}{(h-1)\hat{h}} \zeta^2 + \left( \frac{\gamma}{2} + 6L\gamma^2 \right) \left( 2\gamma_c^2 L^2 (K-1)\sigma^2 + 8\gamma_c^2 L^2 K(K-1)\zeta^2 \right)$$

$$+ \left( \frac{10}{\gamma} \gamma_s^2 + L\gamma_s^2 \right) \mathbb{E}_t \left[ \|e_t\|^2 \right]$$

Taking total expectation from both sides yields the result. $\square$

# B  SUBSAMPLING

In this section, we prove results related to the subsampling process in FedRo. We will first show a few simple properties of the quantity $D(\alpha, \beta)$ as defined in Lemma 1, which will be useful in the subsequent proofs. We will then prove Lemma 1 which provides a sufficient condition for the convergence of FedRo. We then give proofs of Lemma 2, 3, 4, which help us set parameters $\hat{b}$ and $\hat{n}$. In the remaining, let $\hat{b}_t := \left| \hat{\mathcal{B}}_t \right|$.

## B.1  PROPERTIES OF $D(\alpha, \beta)$

We first prove a few simple properties of $D(\alpha, \beta)$ that will be useful in subsequent proofs.

**Property 1.** *For any $\alpha, \beta \in [0, 1]$, we have*

$$\frac{\partial}{\partial \alpha} D(\alpha, \beta) = \ln\left(\frac{\alpha}{\beta}\right) - \ln\left(\frac{1-\alpha}{1-\beta}\right).$$

*Proof.* Note that

$$D(\alpha, \beta) = \alpha \ln\left(\frac{\alpha}{\beta}\right) + (1-\alpha)\ln\left(\frac{1-\alpha}{1-\beta}\right)$$
$$= \alpha \ln(\alpha) + (1-\alpha)\ln(1-\alpha) - \alpha \ln(\beta) - (1-\alpha)\ln(1-\beta).$$

Differentiating this expression with respect to $\alpha$, we get

$$\frac{\partial}{\partial \alpha} D(\alpha, \beta) = \ln(\alpha) + 1 - \ln(1-\alpha) - 1 - \ln(\beta) + \ln(1-\beta)$$
$$= \ln\left(\frac{\alpha}{\beta}\right) - \ln\left(\frac{1-\alpha}{1-\beta}\right).$$

$\square$

**Property 2.** *$D(\alpha, \beta)$ is a decreasing function of the first argument for $\alpha < \beta$, and an increasing function of the first argument if $\alpha > \beta$.*

*Proof.* Suppose $\alpha < \beta$. Then, $\ln(\alpha/\beta) < 0$ and $\ln(1-\alpha/1-\beta) > 0$, hence, the derivative from Property 1 is negative. Case $\alpha > \beta$ follows in a similar fashion. $\square$

**Property 3.** *$D(\alpha, \beta) > 0$ for $\alpha \neq \beta$, and $D(\alpha, \beta) = 0$ for $\alpha = \beta$.*

*Proof.* Observe that $D(\beta, \beta) = \beta \ln(\beta/\beta) + (1-\beta)\ln(1-\beta/1-\beta) = 0$. Now, recall that the derivative with respect to $\alpha$ is negative for all $\alpha < \beta$ and positive for all $\alpha > \beta$. Then, we have $D(\alpha, \beta) > 0$ for any $\alpha \neq \beta$. $\square$

**Property 4.** *$D(\alpha, \beta)$ is convex in the first argument.*

*Proof.* Note that, from Property 1

$$\frac{\partial^2}{\partial \alpha^2} D(\alpha, \beta) = \frac{\partial}{\partial \alpha}\left(\ln\left(\frac{\alpha}{\beta}\right) - \ln\left(\frac{1-\alpha}{1-\beta}\right)\right)$$
$$= \frac{1}{\alpha} + \frac{1}{1-\alpha}$$
$$= \frac{1}{\alpha(1-\alpha)} \geq 4 > 0,$$

since $\alpha(1-\alpha) \leq 1/4$ for any $\alpha \in (0, 1)$. This concludes the proof. $\square$

## B.2 PROOF OF LEMMA 1

For $M, K, m \in \mathbb{N}$ such that $M \geq K, m \geq 0$, consider a population of size $M$ of $K$ distinguished objects, and consider sampling $m$ objects without replacement. Then, the number of distinguished objects in the sample follows a *hypergeometric distribution*, which we denote by $\mathrm{HG}\,(M, K, m)$. Then, $\hat{b}_t \sim \mathrm{HG}\,(n, b, \hat{n})$ for every $t$. By (Chvátal, 1979; Hoeffding, 1994), $\hat{b}_t$ obeys Chernoff bounds. Moreover, we will show that Chernoff bounds are tight for certain regimes of the parameters $M, K, m$ by using binomial approximations of hypergeometric distribution (e.g., Theorem 3.2 of (Holmes, 2004), Theorem 2 of (Ehm, 1991)). These results are summarized in the following lemma.

**Lemma 13.** *Let $X \sim \mathrm{HG}\,(M, K, m)$ be a random variable following a hypergeometric distribution, and let $\beta := {}^K/_M$. Then, for any $1 > \alpha > \beta$, we have*

$$\mathbb{P}\,[X \geq \alpha m] \leq \exp\left(-mD\,(\alpha, \beta)\right),$$

*where $D\,(\alpha, \beta)$ is as in Lemma 1. Moreover, if $\alpha m$ is an integer, then*

$$\mathbb{P}\,[X \geq \alpha m] \geq \frac{1}{\sqrt{8m\alpha(1-\alpha)}} \exp\left(-mD\,(\alpha, \beta)\right) - \frac{m-1}{M-1}.$$

*Proof.* The upper bound is shown in (Chvátal, 1979; Hoeffding, 1994)

$$\mathbb{P}\,[X \geq \alpha m] \leq \exp\left(-mD\,(\alpha, \beta)\right).$$

Now, we show the lower bound. Suppose $\alpha m$ is an integer. Then, Lemma 4.7.1 of (Ash, 2012) gives

$$\mathbb{P}\,[B \geq \alpha m] \geq \frac{1}{\sqrt{8m\alpha(1-\alpha)}} \exp\left(-mD\,(\alpha, \beta)\right), \tag{23}$$

where $B \sim \mathrm{Bin}\,(m, \beta)$ is a binomial random variable. Finally, Theorem 3.2 of (Holmes, 2004) (also Theorem 2 of (Ehm, 1991)) gives

$$|\mathbb{P}\,[B \geq \alpha m] - \mathbb{P}\,[X \geq \alpha m]| \leq \frac{m-1}{M-1}. \tag{24}$$

Combining (23) and (24), we get

$$\mathbb{P}\,[X \geq \alpha m] \geq \mathbb{P}\,[B \geq \alpha m] - \frac{m-1}{M-1}$$

$$\geq \frac{1}{\sqrt{8m\alpha(1-\alpha)}} \exp\left(-mD\,(\alpha, \beta)\right) - \frac{m-1}{M-1},$$

which concludes the proof. $\square$

**Lemma 14.** *Suppose $b$ and $\hat{b}$ are such that ${}^b/_n < {}^{\hat{b}}/_{\hat{n}} < {}^1/_2$. Suppose we have*

$$\hat{n} \geq D\left(\frac{\hat{b}}{\hat{n}}, \frac{b}{n}\right)^{-1} \ln\left(\frac{T}{1-p}\right) \tag{25}$$

*for some $p < 1$. Then for every $t \in \{0, 1, \ldots, T-1\}$, we have*

$$\mathbb{P}\left[\hat{b}_t \geq \hat{b}\right] \leq \frac{1-p}{T}.$$

*Proof.* Recall that $\hat{b}_t \sim \mathrm{HG}\,(n, b, \hat{n})$ follows a hypergeometric distribution. Then, by an upper bound from Lemma 13, we get

$$\mathbb{P}\left[\hat{b}_t \geq \hat{b}\right] \leq \exp\left(-\hat{n}D\left(\frac{\hat{b}}{\hat{n}}, \frac{b}{n}\right)\right).$$

Using (25), we have

$$\mathbb{P}\left[\hat{b}_t \geq \hat{b}\right] \leq \exp\left(-\ln\left(\frac{T}{1-p}\right)\right) = \frac{1-p}{T},$$

which concludes the proof. $\square$

**Lemma 1.** *Let $p < 1$ and $b$ be such that $0 < \raisebox{0.3ex}{$b$}\!/\!\raisebox{-0.3ex}{$n$} < \raisebox{0.3ex}{$1$}\!/\!\raisebox{-0.3ex}{$2$}$. Consider* FedRo *as defined in Algorithm 1. Suppose that $\hat{n}$ and $\hat{b}$ are such that $\raisebox{0.3ex}{$b$}\!/\!\raisebox{-0.3ex}{$n$} < \raisebox{0.3ex}{$\hat{b}$}\!/\!\raisebox{-0.3ex}{$\hat{n}$} < \raisebox{0.3ex}{$1$}\!/\!\raisebox{-0.3ex}{$2$}$ and*

$$\hat{n} \geq \min\left\{ n, D\left(\frac{\hat{b}}{\hat{n}}, \frac{b}{n}\right)^{-1} \ln\left(\frac{T}{1-p}\right) \right\}, \tag{5}$$

*with $D(\alpha, \beta) := \alpha \ln(\raisebox{0.3ex}{$\alpha$}\!/\!\raisebox{-0.3ex}{$\beta$}) + (1 - \alpha) \ln(\raisebox{0.3ex}{$1-\alpha$}\!/\!\raisebox{-0.3ex}{$1-\beta$})$, for $\alpha, \beta \in (0, 1)$. Then, Event $\mathcal{E}$ as defined in (4) holds true with probability at least $p$.*

*Proof.* We will consider two cases: when $\hat{n} = n$, and when $\hat{n} \geq D\left(\raisebox{0.3ex}{$\hat{b}$}\!/\!\raisebox{-0.3ex}{$\hat{n}$}, \raisebox{0.3ex}{$b$}\!/\!\raisebox{-0.3ex}{$n$}\right)^{-1} \ln\left(\raisebox{0.3ex}{$T$}\!/\!\raisebox{-0.3ex}{$1-p$}\right)$.

**(i) Case of $\hat{n} = n$.** By assumption of the lemma $\raisebox{0.3ex}{$b$}\!/\!\raisebox{-0.3ex}{$n$} < \raisebox{0.3ex}{$\hat{b}$}\!/\!\raisebox{-0.3ex}{$\hat{n}$}$, which implies $b < \hat{b}$ since $\hat{n} = n$. Note that we also have $\hat{b}_t = b < \hat{b}$ for all $t \in \{0, 1, \ldots, T - 1\}$, since the entire set of clients is sampled in each round. Then, assertion $\mathcal{E}$ holds with probability $1$.

**(ii) Case of $\hat{n} \geq D\left(\raisebox{0.3ex}{$\hat{b}$}\!/\!\raisebox{-0.3ex}{$\hat{n}$}, \raisebox{0.3ex}{$b$}\!/\!\raisebox{-0.3ex}{$n$}\right)^{-1} \ln\left(\raisebox{0.3ex}{$T$}\!/\!\raisebox{-0.3ex}{$1-p$}\right)$.** Using Lemma 14 and a union bound over all $t \in \{0, 1, \ldots, T - 1\}$, we have

$$\Pr[\neg\mathcal{E}] = \Pr\left[\bigvee_{t\in\{0,1,\ldots,T-1\}} \left\{\hat{b}_t > \hat{b}\right\}\right]$$
$$\leq \sum_{t=0}^{T-1} \Pr\left[\hat{b}_t > \hat{b}\right]$$
$$\leq T \cdot \frac{1-p}{T} = 1 - p.$$

Hence,

$$\Pr[\mathcal{E}] \geq p,$$

which concludes the proof. $\qquad\square$

### B.3  PROOF OF LEMMA 2

**Lemma 2.** *Consider the sampling threshold $\hat{n}_{th}$ defined as follows*

$$\hat{n}_{th} := \left\lceil D\left(\frac{1}{2}, \frac{b}{n}\right)^{-1} \ln\left(\frac{4T}{1-p}\right) \right\rceil + 2, \tag{9}$$

*with $D$ as defined in Lemma 1. If $\hat{n} \geq \hat{n}_{th}$, then there exist $\hat{b}$ such that (5) holds and $\raisebox{0.3ex}{$b$}\!/\!\raisebox{-0.3ex}{$n$} < \raisebox{0.3ex}{$\hat{b}$}\!/\!\raisebox{-0.3ex}{$\hat{n}$} < \raisebox{0.3ex}{$1$}\!/\!\raisebox{-0.3ex}{$2$}$.*

*Proof.* Let $\beta = b/n$, and let $\alpha \in [\raisebox{0.3ex}{$1$}\!/\!\raisebox{-0.3ex}{$2$} - \raisebox{0.3ex}{$1$}\!/\!\raisebox{-0.3ex}{$\hat{n}$}, \raisebox{0.3ex}{$1$}\!/\!\raisebox{-0.3ex}{$2$})$ be arbitrary. We denote by $D_\alpha(\alpha, \beta)$ the derivative of $D(\alpha, \beta)$ with respect to $\alpha$. Recall that, from Property 1, we have

$$D_\alpha(\alpha, \beta) = \frac{\partial}{\partial\alpha} D(\alpha, \beta) = \ln\left(\frac{\alpha}{\beta}\right) - \ln\left(\frac{1-\alpha}{1-\beta}\right)$$

By Property 4, $D(\alpha, \beta)$ is convex in the first argument. Then, by Taylor expansion around $\alpha = 1/2$, we have

$$D(\alpha, \beta) \geq D\left(\frac{1}{2}, \beta\right) + \left(\alpha - \frac{1}{2}\right) D_\alpha \left(\frac{1}{2}, \beta\right)$$

$$= D\left(\frac{1}{2}, \beta\right) - \left(\frac{1}{2} - \alpha\right) D_\alpha \left(\frac{1}{2}, \beta\right)$$

$$= \frac{1}{2} \ln\left(\frac{1/2}{\beta}\right) + \frac{1}{2} \ln\left(\frac{1/2}{1-\beta}\right) - \left(\frac{1}{2} - \alpha\right) \left(\ln\left(\frac{1/2}{\beta}\right) - \ln\left(\frac{1/2}{1-\beta}\right)\right)$$

$$= \alpha \ln\left(\frac{1/2}{\beta}\right) + (1-\alpha) \ln\left(\frac{1/2}{1-\beta}\right)$$

$$= 2\alpha \left(\frac{1}{2}\ln\left(\frac{1/2}{\beta}\right) + \frac{1}{2}\ln\left(\frac{1/2}{1-\beta}\right)\right) + (1-2\alpha)\ln\left(\frac{1/2}{1-\beta}\right)$$

$$= (2\alpha) D\left(\frac{1}{2}, \frac{b}{n}\right) + (1-2\alpha)\ln\left(\frac{1/2}{1-\beta}\right)$$

Since $\beta \in (0,1)$, we have $\ln\left(\frac{1/2}{1-\beta}\right) = \ln\left(\frac{1}{1-\beta}\right) - \ln(2) > -\ln(2)$. Since $\alpha < 1/2$, this implies a strict inequality as follows

$$D(\alpha, \beta) > (2\alpha) D\left(\frac{1}{2}, \frac{b}{n}\right) - (1-2\alpha)\ln(2)$$

Recall that $\alpha \in [1/2 - 1/\hat{n}, 1/2)$. Hence, $\alpha \geq 1/2 - 1/\hat{n}$, and we get

$$D(\alpha, \beta) > \left(1 - \frac{2}{\hat{n}}\right) D\left(\frac{1}{2}, \frac{b}{n}\right) - \frac{2\ln(2)}{\hat{n}}$$

$$\geq \frac{(\hat{n} - 2)D\left(\frac{1}{2}, \frac{b}{n}\right) - 2\ln(2)}{\hat{n}}$$

Since $\hat{n} \geq \hat{n}_{th} \geq D(1/2, b/n)^{-1} \ln(4T/1-p) + 2$, we have

$$D(\alpha, \beta) > \frac{D\left(\frac{1}{2}, \frac{b}{n}\right)^{-1} \ln\left(\frac{4T}{1-p}\right) D\left(\frac{1}{2}, \frac{b}{n}\right) - 2\ln(2)}{\hat{n}}$$

$$= \frac{\ln\left(\frac{4T}{1-p}\right) - 2\ln(2)}{\hat{n}}$$

$$= \frac{\ln\left(\frac{T}{1-p}\right)}{\hat{n}} \tag{26}$$

Note that since $p \geq 0$ and $T \geq 1$, the value on the right hand side is non-negative. Hence, for any $\alpha \in [1/2 - 1/\hat{n}, 1/2)$, we have $D(\alpha, \beta) > 0$. Since $D(\beta, \beta) = 0$, this means that we have $\beta \notin [1/2 - 1/\hat{n}, 1/2)$. Since $\beta < 1/2$, this implies

$$\beta < \frac{1}{2} - \frac{1}{\hat{n}}. \tag{27}$$

Now, set $\hat{b} := \lceil \hat{n}(1/2 - 1/\hat{n}) \rceil$. First, note that $\hat{b} = \lceil \hat{n}(1/2 - 1/\hat{n}) \rceil = \lceil \hat{n}/2 \rceil - 1$. Hence,

$$\hat{n}\left(\frac{1}{2} - \frac{1}{\hat{n}}\right) \leq \hat{b} < \frac{\hat{n}}{2} \tag{28}$$

Hence, $\hat{b}/\hat{n} \in [1/2 - 1/\hat{n}, 1/2)$, and we get from (26)

$$D\left(\frac{\hat{b}}{\hat{n}}, \beta\right) > \frac{\ln\left(\frac{T}{1-p}\right)}{\hat{n}}.$$

Hence, such value of $\hat{b}$ satisfies (5). Using (27) and (28), we also get $b/n < \hat{b}/\hat{n} < 1/2$, which concludes the proof.

$\square$

### B.4 PROOF OF LEMMA 3

**Lemma 15.** *Suppose $0 < \beta < 1/2$, and $\alpha$ is such that $\alpha \in [1/2, 1/2 + 1/2\hat{n}]$ for $\hat{n} \geq 2$. Then*

$$D(\alpha, \beta) \leq \left(1 + \frac{1}{\hat{n}}\right)\left(D\left(\frac{1}{2}, \beta\right) + 1\right).$$

*Proof.* Let $\alpha_0 = 1/2 + 1/2\hat{n}$. Note that

$$D(\alpha, \beta) \leq D(\alpha_0, \beta)$$

$$= \alpha_0 \ln\left(\frac{\alpha_0}{\beta}\right) + (1 - \alpha_0)\ln\left(\frac{1 - \alpha_0}{1 - \beta}\right)$$

$$= \alpha_0 \ln\left(\frac{1/2}{\beta}\right) + (1 - \alpha_0)\ln\left(\frac{1/2}{1 - \beta}\right) + \alpha_0 \ln(2\alpha_0) + (1 - \alpha_0)\ln(2(1 - \alpha_0))$$

$$= 2\alpha_0\left(\frac{1}{2}\ln\left(\frac{1/2}{\beta}\right) + \frac{1}{2}\ln\left(\frac{1/2}{1 - \beta}\right)\right)$$

$$\quad + (1 - 2\alpha_0)\ln\left(\frac{1/2}{1 - \beta}\right) + \alpha_0 \ln(2\alpha_0) + (1 - \alpha_0)\ln(2(1 - \alpha_0))$$

$$= 2\alpha_0 D\left(\frac{1}{2}, \beta\right) + (1 - 2\alpha_0)\ln\left(\frac{1/2}{1 - \beta}\right) + \alpha_0 \ln(2\alpha_0) + (1 - \alpha_0)\ln(2(1 - \alpha_0))$$

$$= 2\alpha_0 D\left(\frac{1}{2}, \beta\right) + (1 - 2\alpha_0)\ln\left(\frac{1/2}{1 - \beta}\right) + \ln(2) - H(\alpha_0),$$

where $H(\alpha_0) = -\alpha_0 \ln(\alpha_0) - (1 - \alpha_0)\ln(1 - \alpha_0)$. Note that $\ln(\alpha_0), \ln(1 - \alpha_0) < 0$, hence $H(\alpha_0) \geq 0$. Therefore, we get

$$D(\alpha, \beta) \leq 2\alpha_0 D\left(\frac{1}{2}, \beta\right) + (1 - 2\alpha_0)\ln\left(\frac{1/2}{1 - \beta}\right) + \ln(2).$$

Note that $\beta \in (0, 1/2)$, hence $\ln\left(\frac{1/2}{1 - \beta}\right) = \ln\left(\frac{1}{1 - \beta}\right) - \ln(2) \geq -\ln(2)$. Also, recall that $(1 - 2\alpha_0) = -\frac{1}{\hat{n}}$. Therefore,

$$D(\alpha, \beta) \leq 2\alpha_0 D\left(\frac{1}{2}, \beta\right) + \frac{1}{\hat{n}}\ln(2) + \ln(2)$$

$$= \left(1 + \frac{1}{\hat{n}}\right)\left(D\left(\frac{1}{2}, \beta\right) + \ln(2)\right)$$

$$\leq \left(1 + \frac{1}{\hat{n}}\right)\left(D\left(\frac{1}{2}, \beta\right) + 1\right),$$

which concludes the proof. $\qquad\square$

**Lemma 3.** *Let $p \geq 1/2$ and suppose that the fraction $b/n < 1/2$ is a constant. Consider* FedRo *as defined in Algorithm 1 with*

$$\hat{n} < \left(D\left(\frac{1}{2}, \frac{b}{n}\right) + 2\right)^{-1}\ln\left(\frac{T}{3(1 - p)}\right) - 1,$$

*where $D$ is defined in Lemma 1. Then for large enough $n$ and any $\hat{b} < \hat{n}/2$, Event $\mathcal{E}$ as defined in (4) holds true with probability strictly smaller than $p$.*

*Proof.* Note that for any $\hat{b} < \hat{n}/2$, we have

$$\Pr[\mathcal{E}] = \Pr\left[\bigwedge_{t \in \{0, 1, \dots, T-1\}}\left(\hat{b}_t \leq \hat{b}\right)\right] \leq \Pr\left[\bigwedge_{t \in \{0, 1, \dots, T-1\}}\left(\hat{b}_t < \frac{\hat{n}}{2}\right)\right]. \tag{29}$$

Recall that $\hat{b}_t \sim \text{HG}(n, b, \hat{n})$. For the duration of this proof, set $\beta := b/n$.

**Case $\hat{n} \geq 2$.** Note that, for any $\hat{n} \in \mathbb{N}$, one of $\hat{n}/2$ and $\hat{n}+1/2$ is an integer. Let $\lambda\hat{n}$ be that integer. Then $\lambda \in [1/2, 1/2 + 1/2\hat{n}]$. Then, by Lemma 15, we get

$$D(\lambda, \beta) \leq \left(1 + \frac{1}{\hat{n}}\right)\left(D\left(\frac{1}{2}, \beta\right) + 1\right). \tag{30}$$

Since $\hat{n} \geq 2$, we also have $\lambda \leq 1/2 + 1/2\hat{n} < 1$. Then, by the lower bound in Lemma 13, we have

$$\mathbb{P}[\hat{b}_t \geq \hat{n}/2] \geq \mathbb{P}[\hat{b}_t \geq \lambda\hat{n}]$$
$$\geq \frac{1}{\sqrt{8\hat{n}\lambda(1-\lambda)}}\exp\left(-\hat{n}D(\lambda, \beta)\right) - \frac{\hat{n}-1}{n-1}$$

Note that $\lambda(1-\lambda) \leq 1/4$, hence, we have

$$\mathbb{P}[\hat{b}_t \geq \hat{n}/2] \geq \frac{1}{\sqrt{2\hat{n}}}\exp\left(-\hat{n}D(\lambda, \beta)\right) - \frac{\hat{n}-1}{n-1}.$$

By (30), we get

$$\mathbb{P}[\hat{b}_t \geq \hat{n}/2] \geq \frac{1}{\sqrt{2\hat{n}}}\exp\left(-(\hat{n}+1)\left(D\left(\frac{1}{2}, \beta\right) + 1\right)\right) - \frac{\hat{n}-1}{n-1}$$
$$\geq \exp\left(-(\hat{n}+1)\left(D\left(\frac{1}{2}, \beta\right) + 1\right) - \frac{1}{2}\ln(2\hat{n})\right) - \frac{\hat{n}-1}{n-1}$$

Using $\ln(2\hat{n}) \leq 2(\hat{n}+1)$, we get

$$\mathbb{P}[\hat{b}_t \geq \hat{n}/2] \geq \exp\left(-(\hat{n}+1)\left(D\left(\frac{1}{2}, \beta\right) + 2\right)\right) - \frac{\hat{n}-1}{n-1}$$

Recall that $\hat{n} < \left(D(1/2, \beta) + 2\right)^{-1}\ln\left(T/3(1-p)\right) - 1$. We then get

$$\mathbb{P}[\hat{b}_t \geq \hat{n}/2] > \exp\left(-\ln\left(\frac{T}{3(1-p)}\right)\right) - \frac{\hat{n}-1}{n-1}$$
$$> 3\frac{1-p}{T} - \frac{\hat{n}-1}{n-1}$$

Since $\hat{n}-1/n-1 \to 0$ as $n \to \infty$, for all large enough $n$, we have

$$\mathbb{P}[\hat{b}_t \geq \hat{n}/2] > 2\frac{1-p}{T}$$
$$= 2\frac{(1-p^{1/T})(1+p^{1/T} + \ldots + p^{(T-1)/T})}{T}$$

Using $p \geq 1/2$, we have

$$> 2\frac{(1-p^{1/T}) \cdot T/2}{T}$$
$$= 1 - p^{1/T}$$

Since sampling is done in an i.i.d. manner, we get

$$\mathbb{P}\left[\bigwedge_{t \in \{0,1,\ldots,T-1\}}\left(\hat{b}_t < \hat{n}/2\right)\right] = \left(1 - \mathbb{P}\left[\hat{b}_1 \geq \hat{n}/2\right]\right)^T$$
$$< \left(1 - (1 - p^{1/T})\right)^T$$
$$< p,$$

which concludes the proof of this case by (29).

**Case** $\hat{n} = 1$**.** We have

$$2 = \hat{n} + 1 < \left( D\left( \frac{1}{2}, \beta \right) + 2 \right)^{-1} \ln \left( \frac{T}{3(1-p)} \right).$$

Hence,

$$\ln \left( \frac{T}{3(1-p)} \right) > 2D\left( \frac{1}{2}, \beta \right) + 4$$

$$= \ln \left( \frac{1/2}{\beta} \right) + \ln \left( \frac{1/2}{1-\beta} \right) + 4$$

$$= -2\ln(2) + \ln \left( \frac{1}{\beta} \right) + \ln \left( \frac{1}{1-\beta} \right) + 4.$$

Since $\ln(2) < 1$ and $\ln(1/1-\beta) > 0$, we have

$$\ln \left( \frac{T}{3(1-p)} \right) > \ln \left( \frac{1}{\beta} \right) + 2.$$

Then

$$\frac{T}{3(1-p)} > \frac{e^2}{\beta},$$

where $e$ is the Euler's number. This implies

$$\beta > \frac{3e^2(1-p)}{T}. \tag{31}$$

Since $\hat{n} = 1$, we have

$$\mathbb{P}\left[ \bigwedge_{t \in \{0,1,\ldots,T-1\}} \left( \hat{b}_t < \hat{n}/2 \right) \right] = \mathbb{P}\left[ \bigwedge_{t \in \{0,1,\ldots,T-1\}} \left( \hat{b}_t = 0 \right) \right]$$

$$= (1-\beta)^T.$$

Using (31), we get

$$\mathbb{P}\left[ \bigwedge_{t \in \{0,1,\ldots,T-1\}} \left( \hat{b}_t < \hat{n}/2 \right) \right] \le \left( 1 - \frac{3e^2(1-p)}{T} \right)^T.$$

Note that $\left( 1 - \frac{3e^2(1-p)}{T} \right)^T$ is an increasing function of $T$ which approaches $\exp(-3e^2(1-p))$ in the limit $T \to \infty$. Hence,

$$\mathbb{P}\left[ \bigwedge_{t \in \{0,1,\ldots,T-1\}} \left( \hat{b}_t < \hat{n}/2 \right) \right] \le \exp(-3e^2(1-p)). \tag{32}$$

Consider a function $g(p) = 3e^2(1-p) + \ln(p)$. We have $g'(p) = \frac{1}{p} - 3e^2 < 0$ since $p \ge 1/2$. Then, for any $p \in [1/2, 1)$, we have $3e^2(1-p) + \ln(p) = g(p) > g(1) = 0$. Hence $-3e^2(1-p) < \ln(p)$, which implies $\exp(-3e^2(1-p)) < p$. Combining this with (32), we get

$$\mathbb{P}\left[ \bigwedge_{t \in \{0,1,\ldots,T-1\}} \left( \hat{b}_t < \frac{\hat{n}}{2} \right) \right] < p,$$

which concludes the proof by (29).

$\square$

### B.5    PROOF OF LEMMA 4

**Lemma 4.** *Suppose we have $0 < \frac{b}{n} < \frac{1}{2}$ and consider the sampling threshold $\hat{n}_{opt}$ defined as follows*

$$\hat{n}_{opt} := \left\lceil \max\left\{ \frac{1}{(1/2 - b/n)^2}, \frac{3}{b/n} \right\} \ln\left(\frac{4T}{1-p}\right) \right\rceil + 2 \ . \tag{10}$$

*If $\hat{n} \geq \hat{n}_{opt}$, then, the solution $\hat{b}_\star$ to (8) exists and satisfies $\hat{b}_\star / \hat{n} \in \mathcal{O}(b/n)$.*

*Proof.* Let $\beta := b/n$, and let $\eta := 1/2 - \beta > 0$. We begin by showing that the value of $\hat{b}_\star$ exists. We will then show that we have $\hat{b}_\star / \hat{n} \in \mathcal{O}(b/n)$.

**(i) Proof of existence of $\hat{b}_\star$.**   Note that

$$D(1/2, \beta) = \frac{1}{2}\ln\left(\frac{1}{2\beta}\right) + \frac{1}{2}\ln\left(\frac{1}{2(1-\beta)}\right)$$
$$= \frac{1}{2}\ln\left(\frac{1}{4\beta(1-\beta)}\right).$$

Recall that $\eta = 1/2 - \beta$. Then, we have

$$D(1/2, \beta) = \frac{1}{2}\ln\left(\frac{1}{4\left(\frac{1}{4} - \eta^2\right)}\right)$$
$$= \frac{1}{2}\ln\left(\frac{1}{(1 - 4\eta^2)}\right)$$
$$= -\frac{1}{2}\ln\left(1 - 4\eta^2\right).$$

Using $-\ln(1-x) \geq x$, we get

$$D(1/2, \beta) \geq \frac{1}{2}4\eta^2$$
$$\geq 2\eta^2.$$

From (10) and the fact that $\eta = 1/2 - \beta$, we have

$$\hat{n}_{opt} \geq \left\lceil \frac{1}{\eta^2}\ln\left(\frac{4T}{1-p}\right) \right\rceil + 2$$
$$\geq \left\lceil \frac{1}{2\eta^2}\ln\left(\frac{4T}{1-p}\right) \right\rceil + 2$$
$$\geq \left\lceil D(1/2, \beta)^{-1}\ln\left(\frac{4T}{1-p}\right) \right\rceil + 2 = \hat{n}_{th}.$$

Hence, the value of $\hat{b}_\star$ exists by Lemma 2.

**(ii) Proof of $\hat{b}_\star / \hat{n} \in \mathcal{O}(b/n)$**

In the rest of the proof, consider two cases.

**(ii.1) Case of $\beta \geq 1/8$.**

Since $\hat{b}_\star$ exists and is upper bounded by $1/2$, we have

$$\frac{\hat{b}_\star}{\hat{n}} \leq \frac{1}{2} \in \mathcal{O}(\beta),$$

where last transition follows by $\beta \geq 1/8$. This concludes the proof of this case.

**(ii.2) Case of $\beta < 1/8$.**

Define $G(\beta) = D(2\beta, \beta)$. Then, we have

$$G(\beta) = 2\beta \ln(2) + (1 - 2\beta) \ln \left( \frac{1 - 2\beta}{1 - \beta} \right).$$

After taking the derivative, we get

$$\frac{\partial}{\partial \beta} G(\beta) = 2 \ln(2) - 2 \ln \left( \frac{1 - 2\beta}{1 - \beta} \right) - \frac{1}{1 - \beta}.$$

Differentiating once again, we have

$$\frac{\partial^2}{\partial \beta^2} G(\beta) = \frac{1}{(1 - \beta)^2 (1 - 2\beta)} > 0,$$

since $\beta < 1/8$. Then, $\frac{\partial}{\partial \beta} G(\beta)$ is an increasing function. Hence, for any $\beta \geq 0$, we have

$$\frac{\partial}{\partial \beta} G(\beta) \geq \frac{\partial}{\partial \beta} G(0) = 2 \ln(2) - 1.$$

Then, since $G(0) = 0$, we have

$$G(\beta) \geq (2 \ln(2) - 1)\beta + G(0) = (2 \ln(2) - 1)\beta > \frac{\beta}{3}.$$

Hence, using the definition of $G$, we get

$$D(2\beta, \beta) > \frac{\beta}{3}.$$

Since $\hat{n}_{opt} \geq 3/\beta \ln(T/1-p)$, we get

$$D(2\beta, \beta) \geq \frac{\ln \left( \frac{T}{1-p} \right)}{\hat{n}_{opt}} \geq \frac{\ln \left( \frac{T}{1-p} \right)}{\hat{n}}. \tag{33}$$

Let $\hat{b} := \lceil 2\beta\hat{n} \rceil$. Then $\hat{b}/\hat{n} \geq 2\beta$. Using Property 2, we have

$$D \left( \frac{\hat{b}}{\hat{n}}, \beta \right) \geq \frac{\ln \left( \frac{T}{1-p} \right)}{\hat{n}}.$$

Hence, $\hat{b}$ satisfies condition (5). Then, by minimality of $\hat{b}_\star$, we have

$$\hat{b}_\star \leq \hat{b} = \lceil 2\beta\hat{n} \rceil \leq 2\beta\hat{n} + 1. \tag{34}$$

From (10) and using $T \geq 1$, $p \geq 0$, we have $\hat{n} \geq \hat{n}_{opt} \geq 3/\beta \ln(4) \geq 3/\beta$. Hence, from (34), we have

$$\frac{\hat{b}_\star}{\hat{n}} \leq 2\beta + \frac{1}{\hat{n}} \leq \frac{7}{3}\beta \in \mathcal{O}(\beta),$$

which concludes the proof. $\square$

## C EXPERIMENTS

In this section we present the full experimental setup of the experiments in Figure 1, 2, 3 and 4.

**Machines used for all the experiments:** 2 NVIDIA A10-24GB GPUs and 8 NVIDIA Titan X Maxwell 16GB GPUs.

## C.1 ADDITIONAL RESULTS ON THE CIFAR-10 DATASET

We evaluate the performance of FedRo on the CIFAR-10 dataset and we highlight the influence of local steps on model accuracy. We consider 150 clients, 15 of which are Byzantine. We use NNM and the Trimmed Mean aggregation rule and run FedRo over 1500 rounds using the Sign Flipping attack and the Fall of Empire attack. The number of subsampled clients $\hat{n}$ and the number of Byzantine clients tolerated per round $\hat{b}$ are determined as in Figure 3, first using Equation (9) to set $\hat{n} = \hat{n}_{th} = 29$, then applying Equation (8) and setting $\hat{b} = 13$. Figure 5 shows the performance of FedRo over 1500 learning rounds. As with the FEMNIST dataset, we see in Figure 6 that the accuracy of the model on the CIFAR-10 dataset increases with the number of local steps performed by clients.

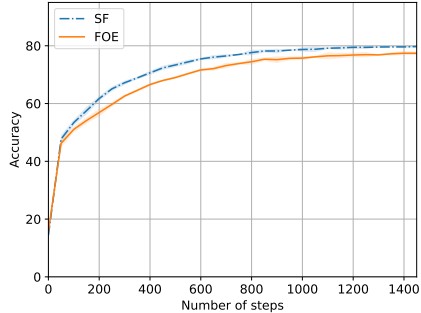

Figure 5: Accuracy of FedRo with NNM and Trimmed Mean on the CIFAR-10 dataset.

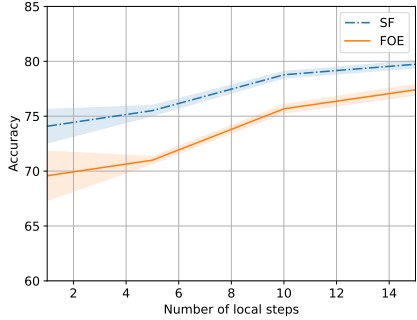

Figure 6: Accuracy of FedRo with respect to the number of local steps on CIFAR-10.

## C.2 FIGURE 1

In this first experiment, we show the variation of $\hat{n}_{th}$ and $\hat{n}_{opt}$ with respect to the fraction of Byzantine workers such that

$$\hat{n}_{th} := \min \left[ n, \left\lceil D\left(\frac{1}{2}, \frac{b}{n}\right)^{-1} \ln\left(\frac{4T}{1-p}\right) \right\rceil + 2 \right] ,$$

and

$$\hat{n}_{opt} := \min \left[ n, \left\lceil \max\left\{ \frac{1}{(1/2 - b/n)^2}, \frac{3}{b/n} \right\} \ln\left(\frac{4T}{1-p}\right) \right\rceil + 2 \right] .$$

We choose $n = 1000$, we set $T = 500$, $p = 0.99$ and make $f/n$ vary in $\left(0, \frac{1}{2}\right)$.

## C.3 FIGURE 2

In the second experiment, we show that there exists an empirical threshold on the number of subsampled clients below which learning is not possible. We use the LEAF Library (Caldas et al., 2018) and download 5% of the FEMNIST dataset distributed in a non-iid way. The exact command line to generate our dataset using the LEAF library is:

```
preprocess.sh -s niid --sf 0.05 -k 0 -t sample
              --smplseed 1549786595 --spltseed 1549786796
```

We summarize the learning hyperparameters in Table 1.

The attack used by Byzantine clients is the sign flipping attack (Allen-Zhu et al., 2020) when the number of sampled Byzantine clients is less than the number of tolerated Byzantines per round $\hat{b}$, otherwise, when the number of sampled Byzantine clients exceeds $\hat{b}$, we consider that they can take control of the learning and set the server parameter to $(0, \ldots, 0) \in \mathbb{R}^d$.

| | |
|---|---|
| Total number of clients | $n = 150$ (first 150 clients of the downloaded dataset) |
| Number of Byzantine clients | $b = 30$ |
| Number of subsampled clients per round | $\hat{n} \in \{1, 6, 11, 16, 21, 26, 31, 36, 41, 46, 51, 56\}$ |
| Number of tolerated Byzantine clients per round | $\hat{b} = \lfloor \frac{\hat{n}}{2} \rfloor - 1$ |
| Model | CNN (architecture presented in Table 2) |
| Algorithm | FedRo |
| Number of steps | $T = 500$ |
| Server step-size | $\gamma_s = 1$ |
| Clients learning rate | $\gamma_c = \begin{cases} 0.1 & \text{if} & 0 \leq T < 300 \\ 0.02 & \text{if} & 300 \leq T < 400 \\ 0.004 & \text{if} & 400 \leq T < 460 \\ 0.0008 & \text{if} & 400 \leq T < 500 \end{cases}$ |
| Loss function | Negative Log Likelihood (NLL) |
| $\ell_2$-regularization term | $10^{-4}$ |
| Aggregation rule | NNM (Allouah et al., 2023) coupled with CW Trimmed Mean (Yin et al., 2018a) |

Table 1: Setup of Figure 2's experiment

| | |
|---|---|
| First Layer | Convolution : in = 1, out = 64, kernel_size = 5, stride = 1 |
| | ReLU |
| | MaxPool(2,2) |
| Second Layer | Convolution : in = 64, out = 128, kernel_size = 5, stride = 1 |
| | ReLU |
| | MaxPool(2,2) |
| Third Layer | Linear : in = $128 \times 4 \times 4$, out = 1024 |
| | ReLU |
| Fourth Layer | Linear : in = 1024, out = 62 |
| | Log Softmax |

Table 2: CNN Architecture

## C.4    FIGURE 3

In this experiments, we show the performance of FedRousing appropriate values for $\hat{n}$ and $\hat{b}$. We use the same portion of the FEMNIST dataset downloaded for the previous experiment (see Section C.3). We list all the hyperparameters used for this experiment in Table 3.

## C.5    FIGURE 4

In this experiments, we show the performance of FedRofor different number of local steps. We use the same portion of the FEMNIST dataset downloaded for the previous experiments (see Section C.3). We list all the hyperparameters used for this experiment in Table 4.

| | |
|---|---|
| Total number of clients | $n = 150$ (first 150 clients of the downloaded dataset) |
| Number of Byzantine clients | $b = 15$ |
| Number of subsampled clients per round | $\hat{n} = 26$ |
| Model | CNN (architecture presented in Table 2) |
| Algorithm | FedRo |
| Number of steps | $T = 500$ |
| Server learning rate | $\gamma_s = 1$ |
| Clients learning rate | $\gamma_c = \begin{cases} 0.1 & \text{if} \quad 0 \leq T < 300 \\ 0.02 & \text{if} \quad 300 \leq T < 400 \\ 0.004 & \text{if} \quad 400 \leq T < 460 \\ 0.0008 & \text{if} \quad 400 \leq T < 500 \end{cases}$ |
| Loss function | Negative Log Likelihood (NLL) |
| $\ell_2$-regularization term | $10^{-4}$ |
| Aggregation rule | NNM (Allouah et al., 2023) coupled with CW Trimmed Mean (Yin et al., 2018a) |
| Byzantine attacks | *sign flipping* (Allen-Zhu et al., 2020), *fall of empires* (Xie et al., 2019), *a little is enough* (Baruch et al., 2019) and *mimic* (Karimireddy et al., 2022) |

Table 3: Setup of Figure 3's experiment

| | |
|---|---|
| Total number of clients | $n = 150$ (first 150 clients of the downloaded dataset) |
| Number of Byzantine clients | $b = 30$ |
| Number of subsampled clients per round | $\hat{n} = 26$ |
| Model | CNN (architecture presented in Table 2) |
| Algorithm | FedRo |
| Number of steps | $T = 500$ |
| Server learning rate | $\gamma_s = 1$ |
| Clients learning rate | $\gamma_c = \begin{cases} 0.1 & \text{if} \quad 0 \leq T < 300 \\ 0.02 & \text{if} \quad 300 \leq T < 400 \\ 0.004 & \text{if} \quad 400 \leq T < 460 \\ 0.0008 & \text{if} \quad 400 \leq T < 500 \end{cases}$ |
| Loss function | Negative Log Likelihood (NLL) |
| $\ell_2$-regularization term | $10^{-4}$ |
| Aggregation rule | NNM (Allouah et al., 2023) coupled with CW Trimmed Mean (Yin et al., 2018a) |
| Byzantine attacks | *sign flipping* (Allen-Zhu et al., 2020), *fall of empires* (Xie et al., 2019), *a little is enough* (Baruch et al., 2019) and *mimic* (Karimireddy et al., 2022) |

Table 4: Setup of Figure 4's experiment

