# OpenReview forum: "Tackling Byzantine Clients in Federated Learning"
_ICLR.cc/2024/Conference — Submitted to ICLR 2024_

### Official Review · Reviewer_xdhf · 2023-10-31

**Soundness:** 2 fair
**Presentation:** 3 good
**Contribution:** 3 good
**Rating:** 6
**Confidence:** 3

**Summary:**

This paper studies the problem of Byzantine clients in federated learning (FL) under client sampling and local steps. The analysis is done under a definition of robustness (specifically, Definition 2) borrowed from reference Allouah et al., 2023 in the paper. A joint condition on the number of sampled clients $\hat{n}$ and maximum number of Byzantine clients $\hat{b}$ (out of $\hat{n}$) is derived such that the robustness condition in Definition 2 is satisfied for all the rounds (with high probability), thereby ensuring convergence. The authors also show that if $\hat{n}$ is too small, then convergence cannot be ensured. Additionally, increasing $\hat{n}$ beyond a certain threshold does not yield any further improvement. Further, it is shown that multiple local steps reduce the asymptotic error. The theoretical claims are corroborated by some experiments.

**Strengths:**

**1.** It appears that this is the first paper analyzing the problem of Byzantine clients with partial-device participation and local steps, and the extension is not trivial. However, I'm not very familiar with related works in this particular area.

**2.** I like the insights in Sections 5.1 and 5.2 namely that if $\hat{n}$ is too small, then convergence cannot be ensured, and increasing $\hat{n}$ beyond a certain threshold does not yield any (order-wise) improvement. (Although I also have some concerns regarding the first insight which I have mentioned in Weakness #1).

**3.** Solid theoretical analysis and the overall presentation is decent.

**Weaknesses:**

**1.** The results in this paper are under the condition in Definition 2 which requires that the number of Byzantine clients in each round (out of $\hat{n}$ total sampled clients) is no more than $\hat{b}$; I think this is a bit stringent. For e.g., it is hard for me to imagine that convergence will suddenly fail instead of becoming slightly worse if there are $\hat{b}+1$ bad clients in only one round (especially if the client updates are bounded). Alternatively, I feel that $\kappa$ in Definition 2 should be a function of $\hat{b}$; for e.g., it is mentioned that $\kappa$ for coordinate-wise trimmed mean is $O(\frac{\hat{b}}{\hat{n}})$. To show the dependence of $\hat{b}$ on $\kappa$, let us denote it as $\kappa(\hat{b})$ instead. In that case, it should be possible to obtain a convergence bound in terms of $\kappa(\hat{b}_1), \kappa(\hat{b}_2),\ldots, \kappa(\hat{b}_T)$ (the subscript denoting the round index).

**2.** The statement of Theorem 1 needs clarification. There is an expectation term in the equation but the line above that says "*with probability at least $p$...*". Is the expectation only over the randomness in stochastic gradients and the randomness in the sampled clients, whereas the probability is over the randomness in the number of Byzantine clients in the set of sampled clients (across all rounds)?

**3.** The lower and upper bound for $\hat{n}$ in Lemma 2 and 4 depend on $T$, but how do you know $T$ in advance in practice?

**4.** In Section 5.2, the authors claim that reference Karimireddy et al., 2021 obtain a matching lower bound of $\Omega(\frac{b}{n}(\frac{\sigma^2}{K} + \zeta^2))$ but looking at their result, it appears that their result is for strongly convex problems and not non-convex problems. So the tightness of Corollary 1 is not clear.

**5.** The $\zeta^2$ term in Corollary 1 doesn't decrease with the number of local steps $K$ and that is usually larger than $\sigma^2$.

**Questions:**

Please see weaknesses.

---

> ### Author Response · Authors · 2023-11-15
> **Response [Part 1]**
>
> > Q: 1. The results in this paper are under the condition in Definition 2 which requires that the number of Byzantine clients in each round (out of $\hat{n}$ total sampled clients) is no more than $\hat{b}$; I think this is a bit stringent. For e.g., it is hard for me to imagine that convergence will suddenly fail instead of becoming slightly worse if there are $\hat{b}+1$ bad clients in only one round (especially if the client updates are bounded). Alternatively, I feel that $\kappa$ in Definition 2 should be a function of $\hat{b}$; for e.g., it is mentioned that $\kappa$ for coordinate-wise trimmed mean is $O({\hat{b}}/{\hat{n}})$. To show the dependence of $\hat{b}$ on $\kappa$, let us denote it as $\kappa(\hat{b})$ instead. In that case, it should be possible to obtain a convergence bound in terms of $\kappa(\hat{b}_1), \kappa(\hat{b}_2), \ldots, \kappa(\hat{b}_T)$.
>
> Re: Indeed, $\kappa$ is a function of $\hat{b}$ for most aggregation rules, including coordinate-wise trimmed mean. However, $\hat{b}$ needs to be set in advance as a parameter for the aggregation rule to effectively tolerate corrupted model updates sent by Byzantine clients.
> When the actual number of Byzantine clients surpasses $\hat{b}$, these clients could arbitrarily manipulate
> the aggregation output, making it impossible (in general) to provide a learning guarantee. Parameter $\hat{b}$ is analogous to
> the breakdown point in the robust statistics literature [C].
>
> To illustrate this,   consider a simple example using the trimmed mean aggregation rule. Suppose we have 5 sampled clients, of which 3 are honest and 2 are Byzantine, with $\hat{b}$ set to 1. If all honest clients report the same value $0$ and the Byzantine clients report some value $x$, the output of the trimmed mean will be ${x}/{3}$. This example demonstrates that when the actual number of Byzantine clients is more than $\hat{b}$, any value in $(-\infty, \infty)$ will be achievable for the Byzantine clients.
>
>
>
> One potential way to mitigate this is to assume a uniform bound on the norm of correct gradients. Adding this assumption bounds the damage caused by the Byzantine clients even if they have full control over the output of the aggregation rule in some rounds. This assumption is, however, not desirable in the optimization literature as it has been shown not to hold in many scenarios [D]. We will add this remark to the camera-ready version of the paper.
>
> [C] P. J. Huber and E. M. Ronchetti. Robust Statistics: Theory and Methods. Wiley Series in Probability and Statistics. John Wiley \& Sons, 2nd edition, 2009.
>
> [D] L. Nguyen, P. Nguyen, M. van Dijk, P. Richtarik, K. Scheinberg and
> M. Takac, SGD and Hogwild! Convergence without the Bounded Gradients Assumption, in International Conference
> on Machine Learning (ICML 2018).
>
> > Q: 2. The statement of Theorem 1 needs clarification. There is an expectation term in the equation but the line above that says "with probability at least $p . . . "$. Is the expectation only over the randomness in stochastic gradients and the randomness in the sampled clients, whereas the probability is over the randomness in the number of Byzantine clients in the set of sampled clients (across all rounds)?
>
> Re: The reviewer is correct regarding the sources of randomness for the high probability bound and the expectation. We will clarify this in the paper.
>
> > Q: 3. The lower and upper bound for $\hat{n}$ in Lemma 2 and 4 depend on $T$, but how do you know $T$ in advance in practice?
>
> Re:
> In our analysis and our current presentation of the results, an upper bound on the number of rounds $T$, should be known to control the probability of failure. If the number of rounds exceeds this upper bound, the probability of failure increases. In practice, researchers may need to make reasonable estimations or use historical data to approximate an appropriate upper bound for $T.$ Alternatively one can interpret the results as computing the probability of failure as a function of $\hat{n}$ and $T$.  This provides a way to assess the impact of different choices of $T$ on the probability of failure, helping practitioners make informed decisions.

---

> > ### Comment · Reviewer_xdhf · 2023-11-22
> > **Reply to Part 1**
> >
> > Thanks for the rebuttal.
> >
> > In their response to Q1, the authors state: "*Indeed, $\kappa$ is a function of $\hat{b}$ for most aggregation rules, including coordinate-wise trimmed mean... When the actual number of Byzantine clients surpasses $\hat{b}$, these clients could arbitrarily manipulate the aggregation output, making it impossible (in general) to provide a learning guarantee.*" -- Well, if $\kappa$ is a function of $\hat{b}$, then the convergence bound in terms of $\kappa(\hat{b}_1), \ldots, \kappa(\hat{b}_T)$ should reflect the impossibility in providing a learning guarantee. That is why I raised this question.
> >
> > Yes, please add the clarification for Q2.

---

> ### Author Response · Authors · 2023-11-15
> **Response [Part 2]**
>
> > Q: 4. In Section 5.2, the authors claim that reference Karimireddy et al., 2021 obtain a matching lower bound of $\Omega\left(\frac{b}{n}\left(\frac{\sigma^2}{K}+\zeta^2\right)\right)$ but looking at their result, it appears that their result is for strongly convex problems and not non-convex problems. So the tightness of Corollary 1 is not clear.
>
> Re: To avoid confusion, note that the lower bound due to $\sigma$ is obtained by Karimireddy et al., 2021 (ICML), and the lower bound due to $\zeta$ is obtained by Karimireddy et al., 2022 (ICLR). As mentioned by the reviewer,
> both these lower bounds have been obtained for the suboptimality of a smooth and $\mu$-strongly function $F$. Specifically,
>
> $$F(x_{out}) - F^* \in \Omega \left(\frac{b}{n} \cdot \frac{\sigma^2}{\mu}\right) \quad \text{and} \quad F(x_{out}) - F^* \in \Omega \left(\frac{b}{n} \cdot \frac{\zeta^2}{\mu}\right),$$
> where $F^*$ is the minimum of $F(x)$.
> But for a $\mu$-strongly function $F$, we have $\left\lVert F(x)\right\rVert^2 \geq 2 \mu (F(x) - F^*)$. Combining this with the above bounds, we obtain the following lower bounds on the norm of the gradient (as done in Theorem III of (Karimireddy et al., 2022)):
> $$\left\lVert \nabla F(x_{out})\right\rVert^2 \in \Omega \left(\frac{b}{n}  {\sigma^2}\right) \quad \text{and} \quad \left\lVert \nabla F(x_{out})\right\rVert^2 \in \Omega \left(\frac{b}{n}  {\zeta^2}\right).$$
> Note that a lower bound on the gradient norm for strongly convex functions also holds true for more general non-convex functions, as strong convexity is just an additional assumption. In other words, if a lower bound holds for a subset (strongly-convex smooth functions), it also holds for the superset (all smooth functions) unless specifically contradicted by other assumptions, which is not the case in our analysis. Also, note that the additional factor of $K$ in the lower bound with respect to $\sigma$ comes from the fact that in FedRo each client computes $K$ local stochastic gradients (in each round) instead of just $1$.
>
> > Q: 5. The $\zeta^2$ term in Corollary 1 doesn't decrease with the number of local steps $K$ and that is usually larger than $\sigma^2$.
>
> Re:
> Unfortunately, the $\zeta^2$ term is unavoidable even without client subsampling and local steps. In fact, the lower bound on $\zeta^2$ is a fundamental result for Byzantine learning that has been discussed in several previous works (Karimireddy et al., 2022; Allouah et al., 2023).

---

> > ### Comment · Reviewer_xdhf · 2023-11-22
> > **Reply to Part 2**
> >
> > Thanks for the clarification regarding Q4. It may be a good idea to add this discussion in the Appendix for the sake of preciseness.

---

> ### Author Response · Authors · 2023-11-22
>
> Thank you very much for the response.
>
> **Regarding your concern about our response to Q1:**
> Sorry for the misunderstanding.
> Let us denote by $\hat{b}_t$ the actual number of sampled Byzantine clients in learning round $t$.
> Integrating $\hat{b}_1, \ldots, \hat{b}_T$ in the convergence bound is actually far from trivial as $\kappa$
> cannot be expressed as a function of the *actual* number of sampled Byzantine clients. It can only be expressed as a function of the *upper bound* on the number of sampled Byzantine clients which is a **known** parameter for the aggregation rule (see e.g., Allouah et al., 2023). This upper bound is denoted by $\hat{b}$ in our paper.
>
>
> To obtain a convergence guarantee, we show (in Lemma 1) that if the number of subsampled clients is large enough, we have $\hat{b}_t \leq \hat{b}$ for all $t \in [T]$ with probability at least $p$.
> This is why i) our results hold in high probability and ii) $\hat{b}_1, \ldots, \hat{b}_T$ do not appear in the convergence rate. We agree that it is an interesting direction to incorporate $\hat{b}_1, \ldots, \hat{b}_T$ in the convergence guarantee. However, we believe it would require stronger assumptions.

---

### Official Review · Reviewer_jgKT · 2023-11-01

**Soundness:** 3 good
**Presentation:** 3 good
**Contribution:** 2 fair
**Rating:** 5
**Confidence:** 2

**Summary:**

The paper studies robust Federated Learning (FL) algorithms for the setting of byzantine clients. In particular, the authors consider FedRo, which is a variant of the standard FedAvg with the simple averaging operation being replaced by a robust averaging mechanism. Then they analyze the complexity of FedRo while taking into account client subsampling and local steps. To circumvent the impossibility of obtaining convergence of FedRo in scenario where the number of sampled clients $\hat{n}$ can be too small (thereby being flooded by a majority of byzantine clients), the authors also obtain a sufficient condition on the client subsampling size and subsequently demonstrate how to set such threshold. Experiments were provided to validate their results.

**Strengths:**

The work is one of the few that consider client subsampling and local steps; the literature for byzantine FL with both of these seems nascent. The results in the paper indeed are better than the latest work of Data & Diggavi (2021) in this specific direction.

**Weaknesses:**

Experiments do not consider other baselines in the literature and are thus very weak.
This work seems like a quite natural (and mechanical) extension from  (Allouah et al., 2023), which had already addressed the harder problem of heterogeneity than client subsampling and local steps considered in this paper on top of the setting. Along this way, perhaps the arising requirement of a sufficient condition on the client subsampling size is interesting (a bit new) and well treated by the authors.

**Questions:**

It would be nice if the authors can elaborate more on the contribution/novelty in view of the comments on the Weaknesses section.

---

> ### Author Response · Authors · 2023-11-15
>
> We thank the reviewer for the comments and provide a detailed response below.
>
> > Q: Experiments do not consider other baselines in the literature and are thus very weak.
>
>
> Re: We will report on more experiments in the camera-ready version. In fact, we have been running new experiments, evaluating FedRo on the CIFAR-10 dataset, that are consistent with the results we had on the FEMNIST dataset. We already included some of these new results in the revised version of the paper which can be found on page 33, Section C.1.
>
> > Q: This work seems like a quite natural  extension from (Allouah et al., 2023), which had already addressed the harder problem of heterogeneity than client subsampling and local steps considered in this paper on top of the setting. Along this way, perhaps the arising requirement of a sufficient condition on the client subsampling size is interesting (a bit new) and well treated by the authors.
>
> Re:  While we indeed rely upon the robustness definition for aggregation rules introduced by Allouah et al., 2023 (as outlined in Definition 2), we believe that our contributions are significantly beyond mechanical extensions.
>  The main contribution of our paper is a novel analysis of (1) client subsampling and (2) local steps within the context of Byzantine-robust federated learning which is applicable to a large number of existing aggregation techniques. Tackling these two challenges is far from trivial. We argue that the existing literature has largely overlooked these two crucial aspects of FL when analyzing robustness, and our convergence guarantee is much tighter than the state-of-the-art solution (Data \& Diggavi, 2021).
>
>
>
> Moreover, our results provide several new insights for the FL community (in the context of Byzantine robustness), including (1) a
> lower bound on the sample size to ensure a learning guarantee, (2) the phenomenon of diminishing return, i.e., the sample size does not improve the asymptotic error (significantly) beyond a certain threshold, and (3) the favorable dependence of the asymptotic error to the number of local steps.
>
>
> We hope that our response addresses your concerns, in which case we sincerely hope that you would raise your score.

---

### Official Review · Reviewer_iQ55 · 2023-11-01

**Soundness:** 4 excellent
**Presentation:** 4 excellent
**Contribution:** 3 good
**Rating:** 8
**Confidence:** 2

**Summary:**

The paper studied the effect of client sampling and local steps in FedRo in dealing with adversarial clients. It theoretically validated the empirical observation of poor performance given a small sampling size, as well as the diminishing gain when the sample size exceeds a threshold.

**Strengths:**

1. The theory and empirical study matches well.
2. It provides a strong understanding and practical guidance in designing robust FL algorithms in dealing with adversarial agents.
3. The paper is well-presented.

**Weaknesses:**

1. Apart from local steps and client sampling, how do communication compression and local data sampling impact FedRo?

**Questions:**

See weaknesses.

---

> ### Author Response · Authors · 2023-11-15
>
> We thank the reviewer for the comments and provide a detailed response below.
>
> > Q: Apart from local steps and client sampling, how do communication compression and local data sampling impact FedRo?
>
> Re:  These are two fundamental aspects.  We discuss each of them below.
>
>
> Communication Compression: Our paper primarily focuses on convergence properties and robustness in the presence of Byzantine clients. While communication compression is indeed an essential aspect of federated learning, its impact is somewhat orthogonal to the core contributions of our study. However, it is important to notice that compression can introduce additional noise in client-server communication, potentially exacerbating the challenges posed by Byzantine clients. This added noise could lead to an increased error rate, affecting the convergence properties. A similar observation has been made in the context of variance-reduced Byzantine learning with communication compression [A].
>
> Data sampling: The analysis we propose in the paper considers a uniform sampling strategy by the honest clients as it is the most common assumption in the literature [B]. In this case, increasing the local sample size decreases the variance of the stochastic noise $\sigma$ leading to a faster convergence.  Nevertheless, exploring different local data sampling strategies and their impact on Byzantine robustness and learning efficiency is an area we plan to explore in the future.
>
> [A] E. Gorbunov, S. Horváth, P. Richtárik, and G. Gidel. Variance reduction is an
> antidote to byzantines: Better rates, weaker assumptions and communication compression as a
> cherry on the top. In The Eleventh International Conference on Learning Representations (ICLR 2023).
>
> [B] X. Li, K. Huang, W. Yang, S. Wang, and Z. Zhang. On the Convergence of FedAvg on non-iid Data. In International Conference on Learning Representations (ICLR 2020).

---

### Official Review · Reviewer_gVud · 2023-11-04

**Soundness:** 3 good
**Presentation:** 4 excellent
**Contribution:** 3 good
**Rating:** 8
**Confidence:** 3

**Summary:**

The authors explore sampling and local update strategies to combat Byzantine clients in a federated setting. In particular, the authors characterize how many clients the central server should subsample if an upper bound of the number of Byzantine clients is known, among with a characterization of how the # of local client updates diminishes error. The authors empirically support their theory via experiments on FEMNIST.

**Strengths:**

1. The authors characterize a sampling strategy for near-convergent FedRO given an upper bound on number of Byzantine clients. The theory uses very most and common assumptions to FL.

2. The authors theoretically demonstrate that increased local # update steps diminishes error in a Byzantine setting. This is a nontrivial and useful conclusion.

3. The authors provide practical theoretical bounds (lower bounds on subsampling size). if the number of Byzantine clients is less than 1/2.

4. Perhaps most importantly, this appears the first Byzantine setup which admits client subsampling and more than one local update.

**Weaknesses:**

1. Though increasing the number of local update steps reduces error, the total error is vanishing even with optimal sampling. I’m wondering if the result of Theorem 1 can be improved perhaps with additional mild assumptions.

2. The paper assumes a Byzantine-defensive aggregation scheme is being used and thus does not propose any truly novel strategy beyond improved sampling (which apparently does not necessarily lead to full convergence).

3. Further empirical corroboration of the proposed theory is likely needed to convince the FL community to seriously explore subsampling strategies.

**Questions:**

See weaknesses. What further assumptions may lead to a fully vanishing error term?

---

> ### Author Response · Authors · 2023-11-15
>
> We thank the reviewer for the comments and provide a detailed response below.
>
> > Q: 1. Though increasing the number of local update steps reduces error, the total error is vanishing even with optimal sampling. I’m wondering if the result of Theorem 1 can be improved perhaps with additional mild assumptions.
>
>
> There seems to be a small typo in the review, and we believe the reviewer meant that the total error is **not** vanishing. Our response below is based on this latter interpretation, but please do let us know if this is not the case.
>
> Re: The suggestion of the reviewer is appealing. Unfortunately, the non-vanishing error due to $\zeta$ is a fundamental lower bound for Byzantine learning even without client subsampling and local steps. It has been discussed in some previous works (Karimireddy et al. 2022; Allouah et al. 2023).
> Some prior research, such as the work by Karimireddy et al. 2022, proposed methods to mitigate this lower bound by changing the assumptions. However, this approach,
> which we believe is applicable to our scheme,  significantly reduces the fraction of tolerable Byzantine clients, often much below $1/2$.
>
> > Q: 2. The paper assumes a Byzantine-defensive aggregation scheme is being used and thus does not propose any truly novel strategy beyond improved sampling.
>
> Re:  The reviewer is correct: we do not propose any new specific aggregation scheme. The contribution of our paper is to precisely characterize
>  the impact of two important and widely used techniques in federated learning. More specifically, the novelty of our paper lies in the detailed analysis of (1) client sampling and (2) local steps in the context of a generic Byzantine-robust federated learning scheme (namely FedRo), which encompasses a large number of aggregation techniques.
> Our results provide several new insights for the FL community (in the context of Byzantine robustness), including (1) a
> lower bound on the sample size to ensure a learning guarantee, (2) the phenomenon of diminishing return, i.e., the sample size does not improve the asymptotic error (significantly) beyond a certain threshold, and (3) the favorable dependence of the asymptotic error to the number of local steps.
>
> > Q: 3. Further empirical corroboration of the proposed theory is likely needed to convince the FL community to seriously explore subsampling strategies.
>
> Re: Indeed, we will add more experiments to the camera-ready version. In fact, we have been running new experiments, evaluating FedRo on the CIFAR-10 dataset, that are consistent with the results we had on the FEMNIST dataset. We already included some of these new results in the revised version of the paper which can be found on page 33, Section C.1.

---

> > ### Comment · Reviewer_gVud · 2023-11-23
> > **Response to authors**
> >
> > For Q1, indeed I meant the total error is *not* vanishing.
> >
> > Thanks for responding to all of my remarks and questions. I am convinced that it is OK that no new specific aggregation scheme is being suggested and consequently will maintain my score. I look forward to further experiments in any later version.

---

### Author Response · Authors · 2023-11-22

Dear Reviewers,

We appreciate your time and effort in reviewing our paper, and we hope that your concerns have been addressed. As the discussion period ends soon, we hope that you will consider participating in an interactive discussion, and we would be happy to discuss any further concerns that you might have regarding our paper.

---

### Meta-Review · Area_Chair_vBXJ · 2023-12-23

**Metareview:**

The authors study distributed optimization in the presence of Byzantine clients. In particular, they consider a method that performs client sampling and local SGD steps. The key result is a theoretical analysis of the method, showing benefit coming from local steps. While some reviewers suggested that this result is nontrivial, I differ. I believe this result actually is expected and easy to obtain. In fact, I knew about it for a couple years but did not publish the results. The idea is clear: with appropriate stepsize, more local SGD steps lead to a better gradient estimator (one with smaller noise). The less noise in the system, the easier it is to tackle Byzantine workers. Indeed, with no noise, we can just perform a majority vote. The next new results is the combination of Byzantine robustness and partial client participation. However, the difficulty of this very interesting setup is entirely avoided since the authors basically assume it away (via assuming that over all iterations, the Byzantine clients form a small enough fraction of the samples clients). I think the really interesting case is the one where Byzantine clients can form a majority in some iterations. However, this difficult case is not studied.

The fact that new aggregation mechanisms are not studied in the paper is completely OK.

It is also OK that the result contains a heterogeneity term which can't be reduced - there are known lower bounds which show that the presence of such a term is unavoidable.

In summary, one of the two main results is rather trivial (albeit new and interesting), and the other is not sufficiently interesting from a technical point of view. I believe this work would be perfectly suited for a journal such as TMLR.

While two reviewers gave the paper a rather high score, the reviews were overly brief, and the scores not sufficiently justified. I therefore did not base my decision on the numerical score, but on my own reading of the paper, and on the reviews and their quality as measure by the justifications given.

**Justification For Why Not Higher Score:**

The paper could potentially be accepted as a poster, but this would be a very borderline decision. I believe the paper should not be accepted though as the results seem rather incremental (some parts are trivial and other complications are avoided altogether by strong assumptions).

**Justification For Why Not Lower Score:**

N/A

---

### Decision · Program_Chairs · 2024-01-16

Reject